EMBO
Molecular Medicine

# The canonical ER stress IRE1α/XBP1 pathway mediates skeletal muscle wasting during pancreatic cancer cachexia

Aniket S Joshi [ID] [1,2], Meiricris Tomaz da Silva [ID] [1,2], Anh Tuan Vuong [ID] [1,2], Bowen Xu [ID] [1,3], Ravi K Singh[1,2] & Ashok Kumar [ID] [1,2 ✉]

## Abstract

Cancer cachexia is a debilitating syndrome characterized by the progressive loss of skeletal muscle mass with or without fat loss. Recent studies have implicated dysregulation of the endoplasmic reticulum (ER) stress-induced unfolded protein response (UPR) pathways in skeletal muscle under various conditions, including cancer. In this study, we demonstrate that the IRE1α/XBP1 branch of the UPR promotes activation of the ubiquitin–proteasome system, autophagy, JAK-STAT3 signaling, and fatty acid metabolism in the skeletal muscle of the KPC mouse model of pancreatic cancer cachexia. Moreover, we show that the IRE1α/XBP1 pathway is a key contributor to muscle wasting. Skeletal muscle-specific deletion of the XBP1 transcription factor significantly attenuates tumor-induced muscle atrophy. Mechanistically, transcriptionally active XBP1 binds to the promoter regions of genes such as *Map1lc3b*, *Fbxo32*, and *Il6*, which encode proteins known to drive muscle proteolysis. Pharmacological inhibition of IRE1α using 4μ8C in KPC tumor-bearing mice attenuates cachexia-associated molecular changes and improves muscle mass and strength. Collectively, our findings suggest that targeting IRE1α/XBP1 pathway may offer a therapeutic strategy to counteract muscle wasting during pancreatic cancer-induced cachexia.

**Keywords** Muscle Wasting; ER Stress; Unfolded Protein Response; JAK-STAT; Fatty Acid Oxidation
**Subject Categories** Cancer; Musculoskeletal System; Organelles

## Introduction

Cachexia is a devastating multifactorial syndrome characterized by unintentional loss of body weight along with a significant decline in physical function, primarily due to the depletion of skeletal muscle mass and adipose tissue. Cachexia is a predominant feature in many types of cancer, such as gastroesophageal, lung, and pancreatic cancer. The prevalence of cachexia is particularly very high in patients with pancreatic ductal adenocarcinoma (PDAC) where it is present in about 60% of newly diagnosed patients and increasing up to 85% at advanced stages (Argiles et al, 2014; Fearon et al, 2011; Fearon et al, 2012; Hendifar et al, 2019; Poulia et al, 2020). PDAC patients with weight loss >10% show intolerance to antineoplastic treatments and have worse survival outcomes (Nemer et al, 2017). While it is now increasingly evidenced that cancer cachexia involves the participation of various molecules such as proinflammatory cytokines, hormones, neuropeptides, and tumor-derived factors, there are still no therapeutic interventions or drugs for the prevention or treatment of cancer cachexia (Poulia et al, 2020).

Muscle wasting due to neoplastic growth, including pancreatic cancer, occurs primarily due to increased protein breakdown and reduced protein synthesis leading to a net loss in the muscle protein content (Fearon et al, 2011; Setiawan et al, 2023; Talbert and Guttridge, 2022; Tisdale, 2009). The ubiquitin–proteasome system (UPS) and autophagy are the major mechanisms, which mediate bulk of muscle protein degradation in catabolic conditions, including during cancer-induced cachexia (Attaix et al, 1999; Attaix et al, 2005; Zhang et al, 2020). Cancer growth also increases the circulating levels of various proinflammatory cytokines and tumor-derived factors which impinge on skeletal muscle to stimulate the activity of the UPS and autophagy (Liu et al, 2024; Setiawan et al, 2023; Talbert and Guttridge, 2022; Zhang et al, 2017). Inflammatory cytokines and tumor-derived molecules also elicit a metabolic shift towards increased fatty acid oxidation in skeletal muscle of tumor-bearing host (Fukawa et al, 2016). However, the molecular underpinning leading to the loss of skeletal muscle mass during cancer cachexia remains less understood.

Skeletal muscle wasting during cancer cachexia involves deregulation of multiple signaling pathways that can be activated by extracellular or intracellular stimuli (Bonetto et al, 2012; Eskiler et al, 2019; Ma et al, 2017; Setiawan et al, 2023; Silva et al, 2015; Tisdale, 2009). Endoplasmic reticulum (ER) is a major cellular organelle involved in protein synthesis, folding, and structural

[1]Institute of Muscle Biology and Cachexia, University of Houston College of Pharmacy, Houston, TX 77204, USA. [2]Department of Pharmacological and Pharmaceutical Sciences, University of Houston College of Pharmacy, Houston, TX 77204, USA. [3]Department of Biomedical Engineering, University of Houston Cullen College of Engineering, Houston, TX, USA. ✉E-mail: akumar43@Central.UH.EDU

maturation. Perturbation in ER homeostasis due to increased load of unfolded/misfolded proteins or in response to other stimuli, such as hypoxia and nutrient deprivation, leads to the activation of a highly organized signaling network, known as unfolded protein response (UPR). The UPR is initiated through the activation of three ER transmembrane proteins, namely inositol-requiring protein 1α (IRE1α), protein kinase R-like endoplasmic reticulum kinase (PERK), and activating transcription factor 6 (ATF6) (Hetz, 2012; Wang and Kaufman, 2014; Wu and Kaufman, 2006). The main function of the UPR is to alleviate ER stress through distinct mechanisms, such as increasing the protein folding capacity within ER, transiently repressing protein synthesis, and targeting unfolded or misfolded proteins for degradation. While UPR is a physiological response that helps restore ER homeostasis, chronic unmitigated ER stress leads to a maladaptive UPR with deleterious consequences. Indeed, prolonged activation of the UPR, observed in various chronic disease states, has been shown to cause inflammation, insulin resistance, cell death, and tissue degeneration (Hetz, 2012; Wu and Kaufman, 2006).

It is now increasingly evidenced that individual arms of the UPR play distinct roles in the regulation of skeletal muscle physiology and pathophysiology (Afroze and Kumar, 2019; Gallot et al, 2019; Joshi et al, 2024b; Roy and Kumar, 2019). Among the three branches, the IRE1α is the most evolutionarily conserved UPR sensor in metazoans and the only UPR pathway in yeast (Le Goupil et al, 2024). In response to ER stress, IRE1α undergoes autophosphorylation, followed by activation of its endoribonuclease domain that cleaves a 26-nucleotide intronic region of X-box protein 1 (XBP1) mRNA, causing a frameshift and translation of its spliced isoform, known as spliced XBP1 (sXBP1). The sXBP1 protein is an active transcription factor that regulates the expression of multiple genes whose products are involved in protein folding and secretion and ER-associated degradation (ERAD) pathway (Le Goupil et al, 2024; Park et al, 2021). Activated IRE1α can also cleave and degrade select mRNAs by a process termed as regulated IRE1α-dependent decay (RIDD), thereby repressing translation of specific proteins (Hollien et al, 2009; Maurel et al, 2014). Recent studies have investigated the physiological role of the IRE1α/XBP1 signaling in skeletal muscle. Myofiber-specific ablation of IRE1α or XBP1 does not have any major effect on post-natal skeletal muscle growth. However, IRE1α/XBP1 signaling promotes skeletal muscle regeneration following acute or chronic injury through cell non-autonomous mechanisms (He et al, 2021; Joshi et al, 2024a; Roy et al, 2021). Moreover, IRE1α/XBP1 signaling promotes myoblast fusion through augmenting the gene expression of various profusion molecules, including myomaker (Joshi et al, 2024b).

Accumulating evidence suggests that the components of the ER stress-induced UPR pathways are activated in skeletal muscle of various mouse models of cancer cachexia (Belcher et al, 2024; Bohnert et al, 2016; Bohnert et al, 2019; Roy and Kumar, 2019). Interestingly, pan-inhibition of ER stress using 4-PBA leads to loss of muscle mass in normal mice and exacerbates skeletal muscle wasting in the Lewis lung carcinoma mouse model of cancer cachexia (Bohnert et al, 2016). However, the role and molecular underpinning by which individual arms of the UPR regulate skeletal muscle mass during pancreatic cancer cachexia remain completely unknown. In this study, we have investigated the role of the IRE1α/XBP1 arm of the UPR in the regulation of skeletal

muscle mass in the KRAS$^{G12D}$ P53$^{R172H}$ Pdx-Cre$^{+/+}$ (KPC) mouse model of pancreatic cancer cachexia (Michaelis et al, 2017). Our results show that the components of the IRE1α/XBP1 arm of the UPR are significantly elevated in skeletal muscle of KPC tumor-bearing mice. Muscle-specific ablation of XBP1 inhibits the activation of proteolytic systems and muscle wasting in KPC tumor-bearing mice. The IRE1α/XBP1 signaling also activates the IL6-JAK-STAT3 signaling and fatty acid metabolism in skeletal muscle of KPC tumor-bearing mice. Furthermore, sXBP1 transcription factor binds to the promoter region of various genes whose products are involved in the regulation of proteolytic systems, pro-inflammatory signaling, and fat metabolism in skeletal muscle. Finally, our results demonstrate that inhibition of the IRE1α endonuclease activity using a small molecule attenuates skeletal muscle wasting in KPC-tumor bearing mice.

# Results

## Activation of IRE1α/XBP1 signaling in skeletal muscle during cancer cachexia

We first determined how the markers of ER stress and UPR are affected in skeletal muscle of mice in response to KPC tumor growth. Specifically, about $2 \times 10^5$ KPC cells were injected into the tail of the pancreas of 12-week-old male mice whereas control mice received an injection of PBS. On day 21 of tumor implantation, there was a marked reduction in body weight and four-paw grip strength of mice injected with KPC cells compared to age-matched PBS-injected control mice (Fig. 1A–C). The mice were finally euthanized, and hind limb muscles were isolated and processed for biochemical analysis. Skeletal muscle wasting during cancer cachexia involves the activation of proteolytic systems, especially ubiquitin–proteasome system (UPS) and autophagy (Argiles et al, 2014). We measured transcript levels of some of the markers of UPS and autophagy in skeletal muscle of control and KPC tumor-bearing mice. Results showed that mRNA levels of muscle-specific E3 ubiquitin ligases, MAFbx (gene name: *Fbxo32*), MuRF1 (gene name: *Trim63*), and MUSA1 (gene name: *Fbxo30*) and autophagy-related molecules, Beclin1 (gene name: *Becn1*), LC3B (gene name: *Map1lc3b*), and *Atg12* were significantly increased in the gastrocnemius (GA) muscle of KPC tumor-bearing mice compared to control mice (Fig. 1D). We next performed bulk RNA-seq analysis of GA muscle of control and KPC tumor-bearing mice and differentially expressed genes (DEGs) in RNA-seq dataset were identified using a threshold of Log2FC ≥ 0.25 and $P$ value < 0.05. We found 1866 DEGs in GA muscle of KPC tumor-bearing mice compared to control mice, of which 783 downregulated and 1083 upregulated (Fig. 1E). Pathway enrichment analysis using Metascape Annotation tool showed that downregulated genes in GA muscle were associated with extracellular matrix organization, skeletal system development, supramolecular fiber organization, cross-linking collagen fibrils, and neuron projection development (Fig. 1F). In contrast, upregulated genes in GA muscle of KPC tumor-bearing mice were associated with the regulation of autophagy, cellular and protein catabolic processes, response to starvation, and response to ER stress (Fig. 1F). Since the role of ER stress-induced pathways in cancer cachexia remains less understood, we focused our analyses on various ER stress-related

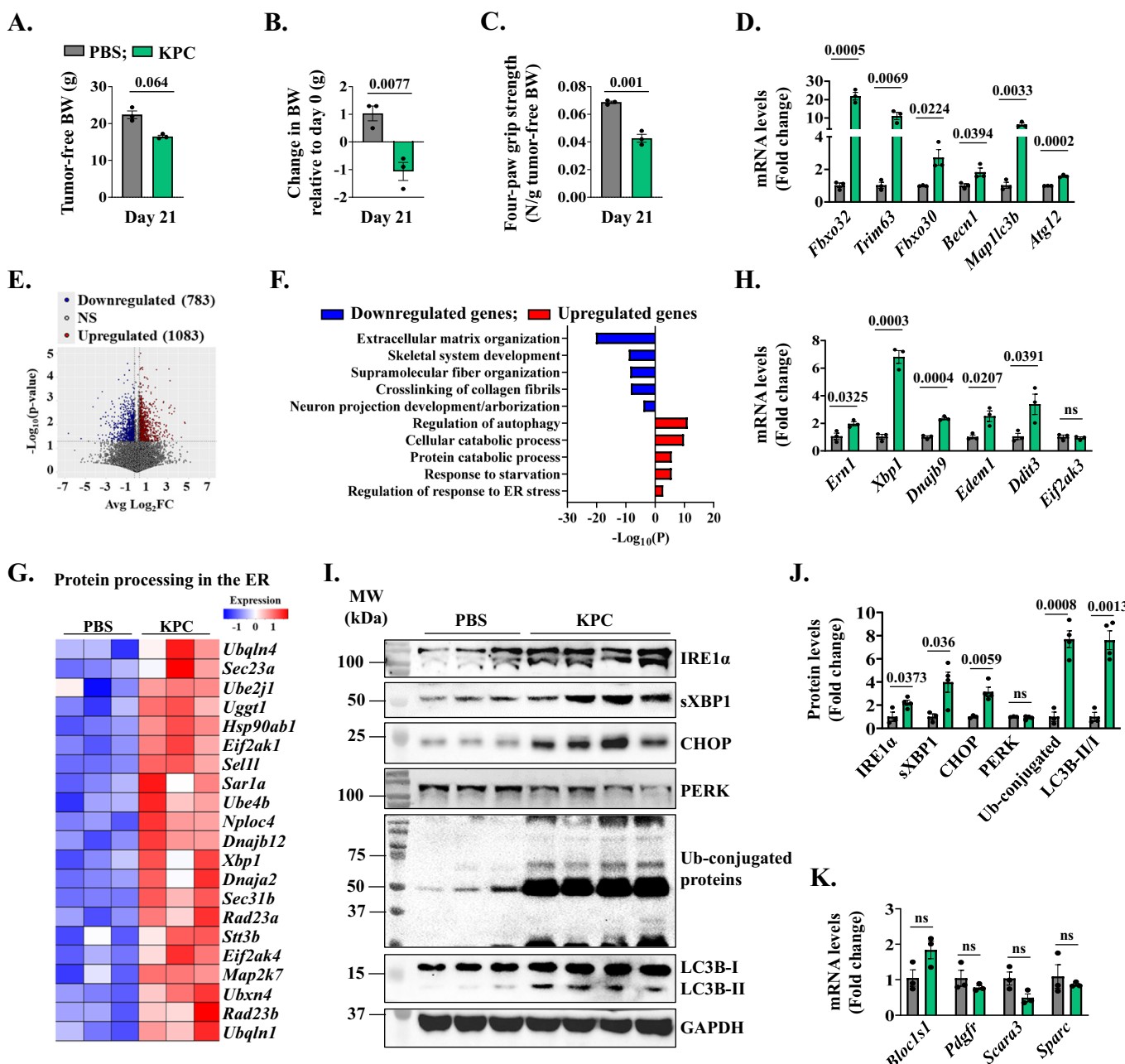

**Figure 1. Activation of IRE1α/XBP1 axis in skeletal muscle of KPC tumor-bearing mice.**

(A–C) Quantification of (A) tumor-free body weight (BW), (B) change in BW relative to day 0, and (C) four-paw grip strength normalized by tumor-free BW on day 21 after injection of KPC cells. $n = 3$ mice per group. Data information: Data are presented as mean ± SEM. Indicated $P$ values were calculated using unpaired Student $t$ test. (D) Relative mRNA levels of MAFbx (*Fbxo32*), MuRF1 (*Trim63*), and MUSA1 (*Fbxo30*), Beclin1 (*Becn1*), LC3b (*Map1lc3b*), and *Atg12* in gastrocnemius (GA) muscle of control and KPC tumor-bearing mice. $n = 3$ mice per group. Data information: Data are presented as mean ± SEM. Indicated $P$ values were calculated using unpaired Student $t$ test. (E) Volcano plot from RNA-Seq dataset analysis presented here shows differentially regulated genes (DEGs) in GA muscle of KPC tumor-bearing mice compared to control mice. $n = 3$ mice per group. Data information: DEGs were identified with the threshold of Log2FC ≥ 0.25 and $P$ value < 0.05 using unpaired Student $t$ test. (F) Bar graph shows pathways associated with downregulated and upregulated mRNAs in GA muscle of KPC tumor-bearing mice compared with control mice identified using Metascape Analysis (metascape.org). (G) Heatmap showing relative mRNA levels of various components involved in protein processing in the ER in GA muscle of control and KPC tumor-bearing mice. $n = 3$ mice per group. Data information: Relative z-scores were calculated from TPM values. (H) Relative mRNA levels of IRE1α (*Ern1*), *Xbp1*, *Dnajb9*, *Edem1*, CHOP (*Ddit3*) and PERK (*Eif2ak3*) in control and KPC tumor-bearing mice. $n = 3$ mice per group. Data information: Data are presented as mean ± SEM. Indicated $P$ values were calculated using unpaired Student $t$ test. (I) Immunoblots and (J) quantification of levels of IRE1α, sXBP1, CHOP, PERK, LC3B and ubiquitin-conjugated proteins in GA muscle of control ($n = 3$) and KPC tumor-bearing mice ($n = 4$ mice). Data information: Data are presented as mean ± SEM. Indicated $P$ values were calculated using unpaired Student $t$ test. (K) Relative mRNA levels of *Bloc1s1*, *Pdgfr*, *Scara3*, and *Sparc* in GA muscle of control and KPC tumor-bearing mice. $n = 3$ mice per group. Data information: Data are presented as mean ± SEM. No significant difference was observed using unpaired Student $t$ test. Source data are available online for this figure.

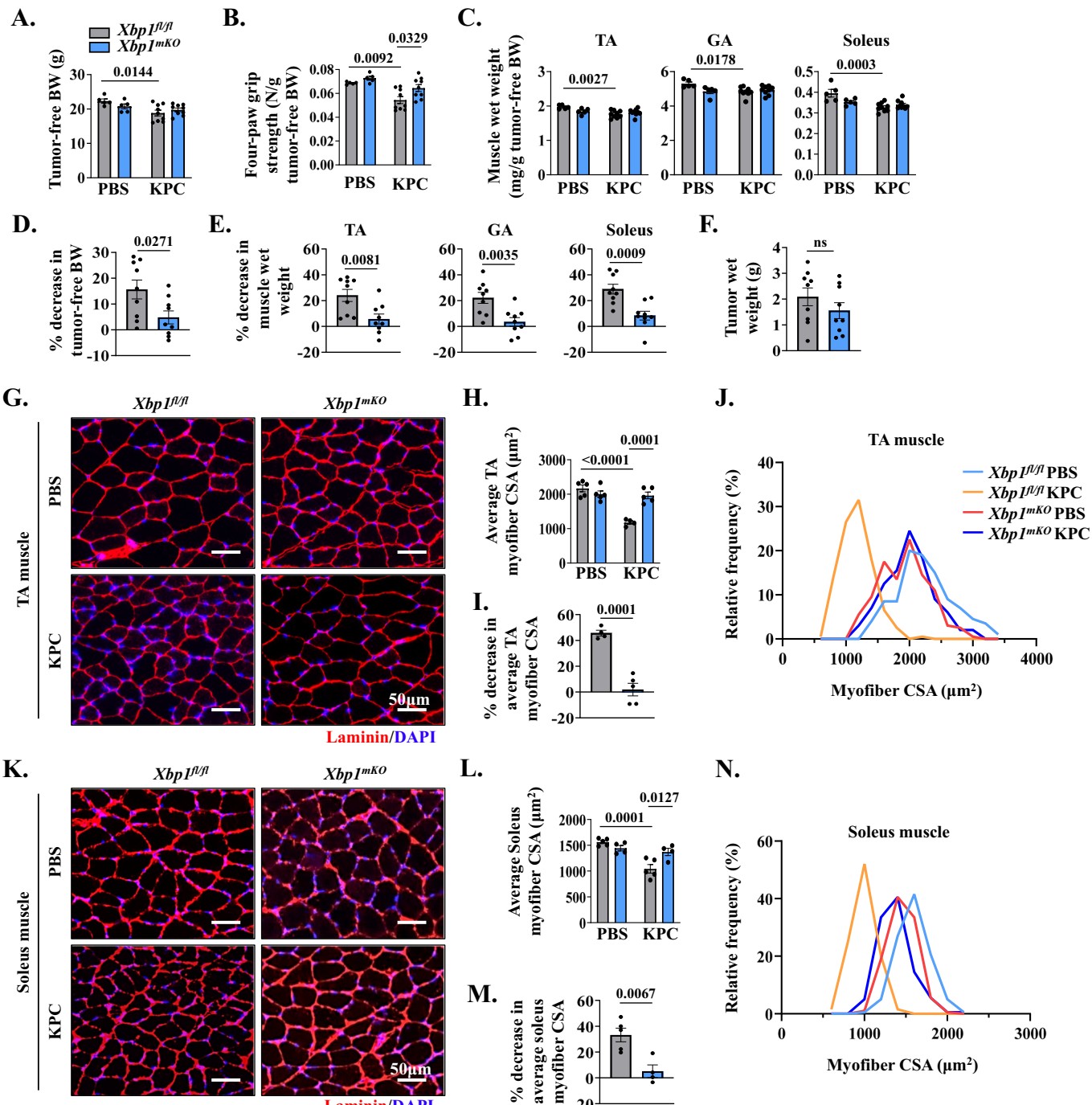

molecules that are affected in skeletal muscle of mice in response to KPC tumor growth. Heatmap analysis of DEGs showed that gene expression of several molecules, especially those involved in the IRE1α/XBP1 arm of the UPR, was highly upregulated in GA muscle of KPC tumor-bearing mice compared to control mice (Fig. 1G). We utilized qRT-PCR to validate increased expression of various transcripts related to the IRE1α/XBP1 arm of the UPR, such as IRE1 (gene name: *Ern1*), *Xbp1*, *Dnajb9*, *Edem1*, and CHOP (gene name: *Ddit3*) in GA muscle of KPC tumor-bearing mice compared to control mice. In contrast, there was no significant difference in

the mRNA levels of PERK (gene name: *Eif2ak3*) between the two groups (Fig. 1H). Western blot analysis showed a significant increase in the protein levels of IRE1α, sXBP1, and CHOP, but not PERK, in GA muscle of KPC tumor-bearing mice compared to controls. There was also a significant increase in the total levels of ubiquitin-conjugated proteins and LC3B-II/I ratio in GA muscle of KPC tumor-bearing mice compared to controls (Fig. 1I,J). To assess autophagy flux in skeletal muscle, we treated control and KPC tumor-bearing mice with colchicine. In the GA muscle of KPC mice, colchicine further increased LC3B-II/I levels, indicating

**Figure 2. Muscle-specific ablation of XBP1 inhibits myofiber atrophy in KPC tumor-bearing mice.**

(A–C) Quantification of (A) tumor-free body weight (BW), and (B) four paw grip strength normalized by tumor-free BW, and (C) Tibialis anterior (TA), gastrocnemius (GA), and soleus muscle wet weight normalized by tumor-free BW. $n = 5$ mice in PBS group and 8–9 mice in KPC group. Data information: Data are presented as mean ± SEM. Indicated $P$ values were calculated using two-way ANOVA followed by Tukey's multiple comparison test. (D, E) Quantification of percentage decrease in (D) tumor-free BW, and (E) TA, GA and soleus muscle wet weight of tumor-bearing $Xbp1^{fl/fl}$ and $Xbp1^{mKO}$ mice compared to corresponding control mice without tumor. $n = 9$ mice per group. Data information: Data are presented as mean ± SEM. Indicated $P$ values were calculated using unpaired Student $t$ test. (F) Quantification of tumor wet weight. $n = 9$ mice per group. Data information: Data are presented as mean ± SEM. No significant difference was observed using unpaired Student $t$ test. (G) Representative photomicrographs of TA muscle sections after anti-laminin staining (to mark myofiber boundaries). DAPI staining was used to identify nuclei. Scale bar, 50 μm. (H) Average myofiber cross-sectional area (CSA) of TA muscle of control and KPC tumor-bearing $Xbp1^{fl/fl}$ and $Xbp1^{mKO}$ mice. $n = 4$–5 mice in each group. Data information: Data are presented as mean ± SEM. Indicated $P$ values were calculated using two-way ANOVA, followed by Tukey's multiple comparison test. (I) Percentage decrease in average CSA, and (J) the distribution of myofiber CSA frequencies in TA muscle of control and KPC tumor-bearing $Xbp1^{fl/fl}$ and $Xbp1^{mKO}$ mice. $n = 4$–5 mice per group. Data information: Data are presented as mean ± SEM. Indicated $P$ values were calculated using unpaired Student $t$ test. (K) Representative photomicrograph of soleus muscle sections after immunostaining for laminin protein and DAPI staining. Scale bar, 50 μm. (L) Average myofiber CSA of soleus muscle of control and KPC tumor-bearing $Xbp1^{fl/fl}$ and $Xbp1^{mKO}$ mice. $n = 4$–5 mice in each group. Data information: Data are presented as mean ± SEM. Indicated $P$ values were calculated using two-way ANOVA followed by Tukey's multiple comparison test. (M) Percentage decrease in average CSA, and (N) the distribution of myofiber CSA frequencies in soleus muscle of control and KPC tumor-bearing $Xbp1^{fl/fl}$ and $Xbp1^{mKO}$ mice. $n = 4$–5 mice per group. Data information: Data are presented as mean ± SEM. Indicated $P$ values were calculated using unpaired Student $t$ test. Source data are available online for this figure.

enhanced autophagy activation in response to tumor burden (Fig. EV1A,B). Next, we performed a puromycin incorporation assay to assess potential differences in the rate of protein synthesis between control and KPC tumor-bearing mice. Intriguingly, skeletal muscles from KPC tumor-bearing mice exhibited a modest but significant increase in puromycin-labeled peptides compared to controls, indicating elevated protein synthesis (Fig. EV1C,D).

To determine whether KPC tumor derived factors are responsible for the upregulation of the IRE1α/XBP1 axis in skeletal muscle, we studied the effect of KPC conditioned medium (KPC-CM) on the levels of IRE1α and sXBP1 in cultured primary myotubes. Results showed a significant upregulation in the levels of IRE1α and sXBP1 in KPC-CM-treated myotubes compared with control myotubes (Fig. EV1E,F). Our results also showed that KPC-CM increases the levels of ubiquitin-conjugated protein and autophagy flux in cultured myotubes (Fig. EV1G–J). We also measured the effect of KPC-CM on the rate of protein synthesis in cultured myotubes. Interestingly, there was a significant upregulation in the amounts of puromycin-tagged protein within 30 min after treatment with KPC-CM. However, at 24 h, puromycin-labeled peptide levels were significantly reduced in KPC-CM–treated cultures compared to controls (Fig. EV1K,L). We next quantified transcript levels of a few RIDD substrates. However, there was no significant difference in the mRNA levels of *Bloc1s1*, *Pdgfr*, *Scara3*, and *Sparc* in GA muscle of control and KPC tumor-bearing mice (Fig. 1K) suggesting that RIDD process is likely not affected in the skeletal muscle of KPC tumor-bearing mice. Altogether, these results suggest that the IRE1α/XBP1 arm of the UPR is activated in skeletal muscle of mice in response to KPC tumor growth.

## Targeted ablation of XBP1 inhibits muscle wasting during pancreatic cancer cachexia

Floxed *Xbp1* (henceforth, $Xbp1^{fl/fl}$) mice were crossed with muscle creatine kinase-Cre (MCK-Cre) mice to generate muscle-specific XBP1 knockout mice (henceforth, $Xbp1^{mKO}$) and littermate $Xbp1^{fl/fl}$ mice. By performing qRT-PCR analysis, we confirmed that there was a significant reduction in the mRNA levels of *Xbp1* exon 2, a sequence flanked by the loxP sites, in $Xbp1^{mKO}$ mice compared to $Xbp1^{fl/fl}$ mice (Fig. EV2A). There was no significant difference in the body weight between male $Xbp1^{fl/fl}$ and $Xbp1^{mKO}$ mice in the

absence of tumor (Fig. EV2B). Next, the pancreas of 12-week-old male $Xbp1^{fl/fl}$ and $Xbp1^{mKO}$ mice was injected with PBS alone or $2 \times 10^5$ KPC cells and the mice were analyzed on day 21 after implantation of KPC cells. Tumor-free body weight of KPC tumor-bearing $Xbp1^{fl/fl}$ mice was significantly reduced compared to corresponding mice injected with PBS alone. In contrast, there was no significant difference in the tumor-free body weight of KPC tumor-bearing $Xbp1^{mKO}$ mice compared to corresponding PBS-injected mice (Fig. 2A). There was also a significant reduction in four-paw grip strength of tumor-bearing $Xbp1^{fl/fl}$ mice compared to corresponding $Xbp1^{fl/fl}$ mice injected with PBS alone. However, no significant difference in grip strength was observed between KPC tumor-bearing $Xbp1^{mKO}$ mice compared to corresponding $Xbp1^{mKO}$ mice without tumor. Indeed, the grip strength of KPC tumor-bearing $Xbp1^{mKO}$ mice was significantly higher compared to KPC tumor-bearing $Xbp1^{fl/fl}$ mice (Fig. 2B). Our analysis also showed that wet weight of tibialis anterior (TA), gastrocnemius (GA), and soleus muscle normalized by tumor-free body weight was significantly reduced in KPC tumor-bearing $Xbp1^{fl/fl}$, but not $Xbp1^{mKO}$ mice, compared to corresponding control mice without tumor (Fig. 2C). Further analysis showed that the percentage loss of tumor-free body weight as well as wet weight of TA, GA, and soleus muscle was significantly less in $Xbp1^{mKO}$ compared to $Xbp1^{fl/fl}$ mice in response to KPC tumor growth (Fig. 2D,E). However, there was no significant difference in the wet weight of tumors between $Xbp1^{fl/fl}$ and $Xbp1^{mKO}$ mice, suggesting that muscle-specific ablation of XBP1 does not affect KPC tumor growth in mice (Fig. 2F).

To understand the effect of targeted deletion of XBP1 on KPC tumor-induced muscle atrophy, we next generated transverse sections of TA (fast/glycolytic) and soleus (slow/oxidative) muscle, performed anti-laminin and DAPI staining or H&E staining, followed by morphometric analysis (Figs. 2G–N and EV2C–E). There was a significant reduction in the average myofiber cross-sectional area (CSA) of TA and soleus muscle of KPC tumor-bearing $Xbp1^{fl/fl}$ mice, but not $Xbp1^{mKO}$, mice compared to corresponding control mice injected with PBS alone. Interestingly, the average myofiber CSA of TA and soleus muscle of KPC tumor-bearing $Xbp1^{mKO}$ mice was significantly higher compared with corresponding muscle of KPC tumor-bearing $Xbp1^{fl/fl}$ mice (Fig. 2H,L). Our analysis also showed that the percentage decrease in average myofiber CSA in response to KPC tumor growth was

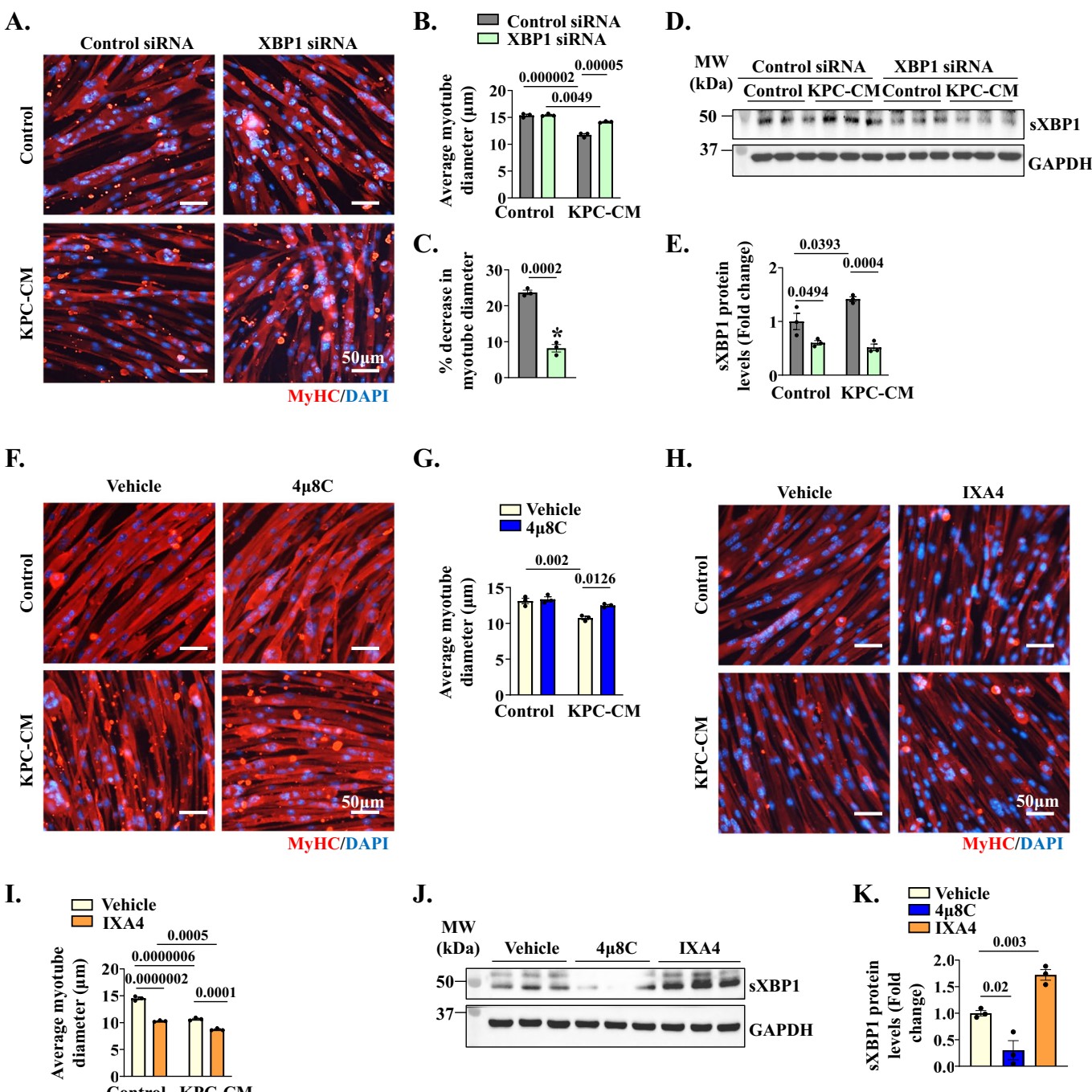

significantly less in TA and soleus muscle of *Xbp1^{mKO}* mice compared with *Xbp1^{fl/fl}* mice (Fig. 2I,M). Frequency distribution analysis of TA and soleus myofiber CSA further showed an increase in the proportion of smaller myofibers in KPC tumor-bearing *Xbp1^{fl/fl}* mice, but not in *Xbp1^{mKO}* mice, compared to corresponding control mice without tumor (Fig. 2J,N). Sirius red staining showed that there was no apparent difference in the collagen deposition in soleus muscle of KPC tumor-bearing *Xbp1^{fl/fl}* and *Xbp1^{mKO}* mice (Fig. EV2F). These results suggest that myofiber-specific ablation of XBP1 inhibits the loss of skeletal muscle mass and strength during pancreatic cancer-induced cachexia.

## IRE1α/XBP1 signaling induces atrophy in cultured myotubes

To further understand the role of IRE1α/XBP1 axis in pancreatic cancer-induced cachexia, we utilized an in vitro model in which cultured myotubes were treated with KPC cells conditioned media (KPC-CM) for 24 h, followed by analysis of myotube diameter. Specifically, cultured mouse primary myotubes were transfected with control or XBP1 siRNA for 24 h followed by treatment with KPC-CM for an additional 24 h. The cultures were fixed and immunostained for myosin heavy chain (MyHC) protein and

**Figure 3.    Role of IRE1α/XBP1 signaling in myotube atrophy in response to KPC tumor-derived factors.**

(A) Representative photomicrographs of control and XBP1 siRNA transfected myotubes incubated with or without KPC-CM. Scale bar, 50 µm. (B) Quantification of average myotube diameter. $n = 3$ (biological replicates) per group. Data information: Data are presented as mean ± SEM. Indicated $P$ values were calculated using two-way ANOVA followed by Tukey's multiple comparison test. (C) Percentage reduction in average myotube diameter in cultures transfected with control or XBP1 siRNA and incubated with or without KPC-CM. $n = 3$ (biological replicates) per group. Data information: Data are presented as mean ± SEM. Indicated $P$ values were calculated using unpaired Student $t$ test. (D) Immunoblot and (E) densitometry analysis of protein levels of sXBP1 in control or XBP1 siRNA transfected myotubes incubated with or without KPC-CM. $n = 3$ biological replicates per group. Data are presented as mean ± SEM. Indicated $P$ values were calculated using two-way ANOVA followed by Tukey's multiple comparison test. (F) Representative photomicrographs of vehicle- or 4 µM 4µ8C-treated myotubes incubated in DM or KPC-CM are presented here. Scale bar, 50 µm. (G) Quantification of average myotube diameter in vehicle alone or 4µ8C-treated cultures incubated with or without KPC-CM. $n = 3$ (biological replicates) per group. Data information: Data are presented as mean ± SEM. Indicated $P$ values were calculated using two-way ANOVA followed by Tukey's multiple comparison test. (H) Representative photomicrographs of vehicle- or 20 µM IXA4-treated myotubes incubated in DM or KPC-CM are presented here. Scale bar, 50 µm. (I) Quantification of average myotube diameter. $n = 3$ biological replicates per group. Data are presented as mean ± SEM. Indicated $P$ values were calculated using two-way ANOVA followed by Tukey's multiple comparison test. (J) Immunoblot and (K) densitometry analysis of protein levels of sXBP1 in 4µ8C or IXA4-treated myotubes compared to control vehicle-treated cultures. $n = 3$ biological replicates per group. Data are presented as mean ± SEM. Indicated $P$ values were calculated using unpaired Student $t$ test. Source data are available online for this figure.

average myotube diameter was measured. There was no significant difference in the average myotube diameter in control or XBP1 siRNA transfected cultures without KPC-CM. However, treatment with KPC-CM significantly reduced the average myotube diameter in both control and XBP1 siRNA-transfected cultures. However, the average myotube diameter was significantly higher in XBP1 siRNA-transfected cultures in comparison to control siRNA transfected cultures when treated with KPC-CM (Fig. 3A–C). Western blot analysis showed significant reduction in protein levels of sXBP1 in cultures transfected with XBP1 siRNAs compared to cultures transfected with control siRNA (Fig. 3D,E).

Although our in vivo and in vitro results suggest that XBP1 is involved in the etiology of pancreatic cancer-induced cachexia, these genetic models reduce the levels of both unspliced and spliced XBP1. To determine whether IRE1α-mediated splicing of XBP1 mRNA is required for myotube atrophy, we studied the effect of 4µ8C, a potent inhibitor of IRE1α RNase activity (Hetz et al, 2019). For this experiment, cultured myotubes were pretreated with vehicle alone or 4µ8C for 1 h followed by addition of KPC-CM. After 24 h, the cultures were analyzed by performing immunostaining for MyHC protein. There was no significant difference in average myotube diameter in vehicle- and 4µ8C-treated cultures incubated without any conditioned medium. However, average myotube diameter was significantly higher in 4µ8C-treated cultures compared with vehicle-treated cultures in response to KPC-CM (Fig. 3F,G).

Recently, a small molecule named IXA4 was identified as a highly specific and potent activator of IRE1α endoribonuclease activity (Grandjean et al, 2020). We investigated the effects of IXA4 on average diameter of cultured myotubes. Interestingly, treatment with IXA4 alone led to a drastic reduction in myotube diameter which was further reduced by co-treatment with KPC-CM (Fig. 3H,I). Using western blot analysis, we confirmed that 4µ8C significantly reduced the levels of sXBP1 protein, whereas treatment with IXA4 compound significantly increased the levels of sXBP1 protein in cultured myotubes (Fig. 3J,K). Collectively, these results suggest that IRE1α induces muscle atrophy in response to pancreatic cancer-derived factors through augmenting the levels of sXBP1 protein both in vivo and in vitro.

## Targeted ablation of XBP1 inhibits proteolytic systems in skeletal muscle

To understand the mechanisms by which XBP1 promotes pancreatic cancer-induced muscle wasting, we performed bulk

RNA-seq analysis of GA muscle isolated from $Xbp1^{fl/fl}$ and $Xbp1^{mKO}$ mice on day 21 after injection of PBS or KPC cells into the pancreas. Analysis of DEGs, using the threshold of Log2FC ≥ 0.25 and $P$ value < 0.05, revealed that expression of 152 genes was upregulated, whereas the expression of 1558 genes was down-regulated in GA muscle of KPC tumor-bearing $Xbp1^{mKO}$ mice compared to $Xbp1^{fl/fl}$ mice. Functional enrichment analysis of DEGs showed that the downregulated genes were associated with intracellular protein transport, regulation of cellular catabolic process, proteolysis, and AMPK signaling pathway. In contrast, upregulated genes were involved in the regulation of hormone metabolic process, negative regulation of cell development, response to TGFβ, and negative regulation of protein ubiquitina-tion (Fig. 4A). Heatmap representation of DEGs showed mRNA levels of multiple molecules involved in ubiquitin proteasome system (UPS) and autophagy/mitophagy are significantly increased in GA muscle of KPC tumor-bearing $Xbp1^{fl/fl}$ mice compared to $Xbp1^{fl/fl}$ mice without tumor. Interestingly, muscle-specific ablation of XBP1 diminished the mRNA levels of multiple molecules of UPS or autophagy in KPC tumor-bearing mice (Fig. EV3A,B). Independent qRT-PCR analysis further confirmed that mRNA levels of E3 ubiquitin ligases, MAFbx (gene name: $Fbxo32$) and MuRF1 (gene name: $Trim63$), autophagy markers, LC3b (gene name: $Map1lc3b$) and Beclin1 (gene name: $Becn1$) were significantly reduced in GA muscle of KPC tumor-bearing $Xbp1^{mKO}$ mice compared with KPC tumor-bearing $Xbp1^{fl/fl}$ mice (Fig. 4B). Furthermore, mRNA levels of $Xbp1$ and its known transcriptional target $Dnajb9$ were reduced in GA muscle of KPC tumor-bearing $Xbp1^{mKO}$ mice compared with GA muscle of KPC tumor-bearing $Xbp1^{fl/fl}$ mice (Fig. 4B). Western blot analysis also showed that the levels of total ubiquitin (Ub)-conjugated proteins, MAFbx, MuRF1, LC3b-II, and sXBP1 protein were significantly reduced in GA muscle of KPC tumor-bearing $Xbp1^{mKO}$ mice compared with corresponding $Xbp1^{fl/fl}$ mice (Fig. 4C,D). Consistent with the in vivo results, we found that treatment with 4µ8C also reduced the levels of total Ub-conjugated proteins in cultured myotubes in response to treatment with KPC-CM (Fig. 4E,F).

We also investigated whether the targeted deletion of XBP1 has any impact on other UPR components in skeletal muscle of mice. Results showed that the levels of IRE1α were higher in skeletal muscle of control (i.e., PBS injected) as well as KPC tumor-bearing $Xbp1^{mKO}$ mice compared to corresponding $Xbp1^{fl/fl}$ mice which could be a compensatory mechanism. There was no significant

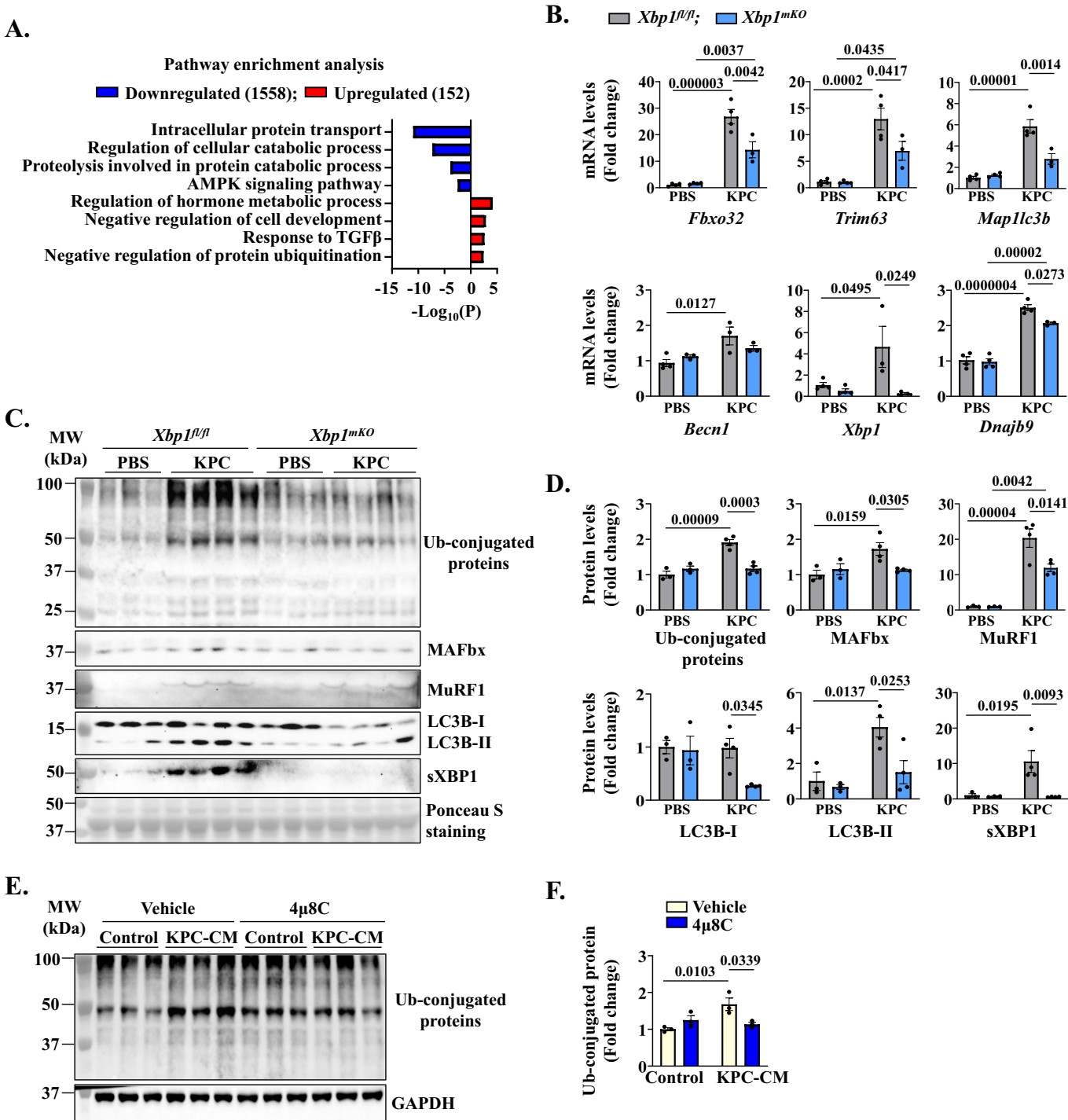

**Figure 4. XBP1 promotes the activation of proteolytic systems in skeletal muscle.**

(A) Bar graph presented here demonstrates enriched pathways associated with downregulated and upregulated mRNAs in KPC tumor-bearing *Xbp1^mKO* mice compared with KPC tumor-bearing *Xbp1^fl/fl* mice identified using Metascape Analysis (metascape.org). (B) Relative mRNA levels of *Fbxo32*, *Trim63*, *Map1lc3b*, *Becn1*, *Xbp1*, and *Dnajb9* in GA muscle of control or KPC tumor-bearing *Xbp1^fl/fl* and *Xbp1^mKO* mice. n = 3–4 mice in each group. Data information: Data are presented as mean ± SEM. Indicated *P* values were calculated using two-way ANOVA followed by Tukey's multiple comparison test. (C) Immunoblots and (D) densitometry analysis showing total levels of ubiquitin-conjugated proteins and MAFbx, MuRF1, LC3B-II/I, and sXBP1 protein in GA muscle of control or KPC tumor-bearing *Xbp1^fl/fl* and *Xbp1^mKO* mice. n = 3–4 mice per group. Data are presented as mean ± SEM. Indicated *P* values were calculated using two-way ANOVA followed by Tukey's multiple comparison test. (E) Immunoblot and (F) quantification analysis showing levels of ubiquitinated proteins in vehicle- and (4 μM) 4μ8C-treated cultures incubated with or without KPC-CM. n = 3 biological replicates per group. Data are presented as mean ± SEM. Indicated *P* values were calculated using two-way ANOVA followed by Tukey's multiple comparison test. Source data are available online for this figure.

difference in the levels of PERK protein between PBS- or KPC cells-injected $Xbp1^{fl/fl}$ and $Xbp1^{mKO}$ mice. Protein levels of CHOP were significantly increased in GA muscle of KPC tumor-bearing $Xbp1^{fl/fl}$ mice, but not in $Xbp1^{mKO}$ mice, compared to corresponding control mice injected with PBS alone (Fig. EV4A,B). ERAD, a crucial process responsible for eliminating misfolded or terminally unfolded proteins through UPS and autophagy, plays an important role in the maintenance of skeletal muscle mass and systemic energy metabolism (Abdon et al, 2023). In the conditions of ER stress, the IRE1α/XBP1 axis augments the gene expression of various molecules that are involved in the ERAD (Hetz, 2012; Le Goupil et al, 2024). Interestingly, our RNA-seq analysis showed that multiple ERAD-related molecules, including Sel1l/Hrd1, were upregulated in the skeletal muscle of KPC tumor-bearing $Xbp1^{fl/fl}$ mice. Moreover, the gene expression of many ERAD-related molecules is significantly reduced in skeletal muscle of KPC tumor-bearing $Xbp1^{mKO}$ mice compared with corresponding $Xbp1^{fl/fl}$ mice (Supplemental Fig. EV4C) suggesting that XBP1-mediated ERAD activation may also contribute to muscle proteolysis during pancreatic cancer cachexia. Collectively, these results suggest that the IRE1α/XBP1 signaling stimulates the activation of proteolytic systems in skeletal muscle during pancreatic cancer cachexia.

## IRE1α/XBP1 axis regulates STAT3 signaling in skeletal muscle during cancer cachexia

The IL6-JAK-STAT3 signaling has been shown to promote skeletal muscle wasting during cancer cachexia (Bonetto et al, 2012; Eskiler et al, 2019; Rupert et al, 2021). The Gene Set Enrichment Analysis (GSEA) of the RNA-seq dataset showed that the expression of many molecules related to IL6-JAK-STAT3 signaling were highly upregulated in GA muscle of KPC tumor-bearing $Xbp1^{fl/fl}$ mice compared to control $Xbp1^{fl/fl}$ mice without tumor (Fig. 5A). Heatmap representation further revealed that the gene expression of various components of the JAK-STAT signaling pathway, was upregulated in the GA muscle of KPC tumor-bearing $Xbp1^{fl/fl}$ mice compared with PBS-injected $Xbp1^{fl/fl}$ mice. Importantly, the gene expression of many JAK-STAT pathway-related molecules is significantly reduced in KPC tumor-bearing $Xbp1^{mKO}$ mice compared with corresponding $Xbp1^{fl/fl}$ mice (Fig. 5B). Consistent with the RNA-seq results, western blot analysis showed that the levels of phosphorylated STAT3 (p-STAT3) and total STAT3 protein were significantly increased in GA muscle of KPC tumor-bearing $Xbp1^{fl/fl}$ mice. Notably, the levels of p-STAT3 and total STAT3 were significantly reduced in KPC tumor-bearing $Xbp1^{mKO}$ mice compared to corresponding $Xbp1^{fl/fl}$ mice suggesting that the IRE1α/XBP1 signaling mediates the activation of the JAK-STAT signaling in skeletal muscle of KPC tumor-bearing mice (Fig. 5C,D).

To validate the role of IRE1α/sXBP1 axis in the activation of STAT3 signaling, we investigated the effect of pharmacological activation and inhibition of IRE1α/XBP1 signaling on the levels of p-STAT3 in cultured myotubes. Primary myotubes were treated with IXA4 (an activator of IRE1α endonuclease activity) for 24 h followed by performing western blot analysis. Results showed that treatment with IXA4 significantly increased the levels of p-STAT3 protein (Fig. 5E,F). In a parallel experiment, cultured myotubes were treated with 4μ8C (an inhibitor of IRE1α endonuclease activity) for 24 h followed by measuring levels of p-STAT3 protein. Results showed that treatment with 4μ8C significantly reduced the

levels of p-STAT3 in cultured myotubes (Fig. 5G,H). Since IXA4 treatment alone is sufficient to reduce myotube size (Fig. 3H) and induce phosphorylation of STAT3, we next sought to determine whether silencing of STAT3 can improve myotube diameter in IXA4-treated cultures. For this experiment, primary myotubes were transfected with control or STAT3 siRNA and incubated with or without IXA4 for 24 h. The cultures were then fixed and immunostained for MyHC protein (Fig. 5I). Our analysis showed that the knockdown of STAT3 significantly increased the average myotube diameter in IXA4-treated cultures (Fig. 5J). By performing western blot, we confirmed that protein levels of STAT3 were significantly reduced in cultures transfected with STAT3 siRNA. Intriguingly, the knockdown of STAT3 also reduced the levels of sXBP1 protein in IXA4-treated cultures (Fig. 5K,L), indicating a feedback mechanism between sXBP1 and STAT3. Collectively, these results suggest that IRE1α/XBP1 signaling promotes muscle wasting through the activation of STAT3 signaling.

## Targeted deletion of XBP1 inhibits fatty acid metabolism in cachectic muscle

It has been reported that an acute increase in circulating levels of IL-6 activates STAT3 signaling and selectively promotes lipid metabolism in human skeletal muscle (Wolsk et al, 2010). We investigated whether IRE1α/XBP1 axis has any role in fatty acid oxidation in skeletal muscle of KPC tumor-bearing mice. The GSEA of RNA-seq dataset showed that the fatty acid metabolism was significantly increased in GA muscle of KPC tumor-bearing mice compared with control mice without tumor (Fig. 6A). Furthermore, the heatmap generated from RNA-seq analysis showed that gene expression of multiple molecules involved in fatty acid metabolism was elevated in GA muscle of KPC tumor-bearing $Xbp1^{fl/fl}$ mice. Interestingly, the mRNA levels of many molecules related to fatty acid metabolism were reduced in GA muscle of KPC tumor-bearing $Xbp1^{mKO}$ mice compared with corresponding muscle of KPC tumor-bearing $Xbp1^{fl/fl}$ mice (Fig. 6B). Our analysis of potential transcriptional regulators of the down-regulated molecules related to fatty acid metabolism using TRRUST database revealed PPARα as one of the regulators (Fig. 6C). PPAR factors, including PPARα, PPARδ, and PPARγ, regulate lipid metabolism in various tissues, including skeletal muscle (Berger and Moller, 2002; Burri et al, 2010; Crossland et al, 2021). In addition, activation of PPARα and PPARδ is linked with increased fatty acid oxidation in skeletal muscle of mice and human subjects (Burri et al, 2010; Crossland et al, 2021). By performing qRT-PCR, we confirmed that mRNA levels of many molecules involved in fatty acid metabolism, including PPARδ (gene name: Ppard), Pgc1α (gene name: Ppargc1a), Cd36, Acox1, Acox2, Acox3, Sirt1, Hif1a, but not Hadhb, were significantly increased in GA muscle of KPC tumor-bearing $Xbp1^{fl/fl}$ mice compared to corresponding control $Xbp1^{fl/fl}$ mice. Importantly, the mRNA levels of most of these molecules were significantly reduced in GA muscle of KPC tumor-bearing $Xbp1^{mKO}$ mice compared to corresponding $Xbp1^{fl/fl}$ mice. Irisin (gene name: Fndc5), a novel myokine that positively regulates muscle mass (Reza et al, 2017), was recently shown to ameliorate chronic kidney disease-induced muscle wasting through the inhibition of fatty acid metabolism (Zhou et al, 2023). Our results demonstrate that mRNA levels of Irisin were significantly reduced in GA muscle of KPC tumor-bearing $Xbp1^{fl/fl}$ mice, but not in

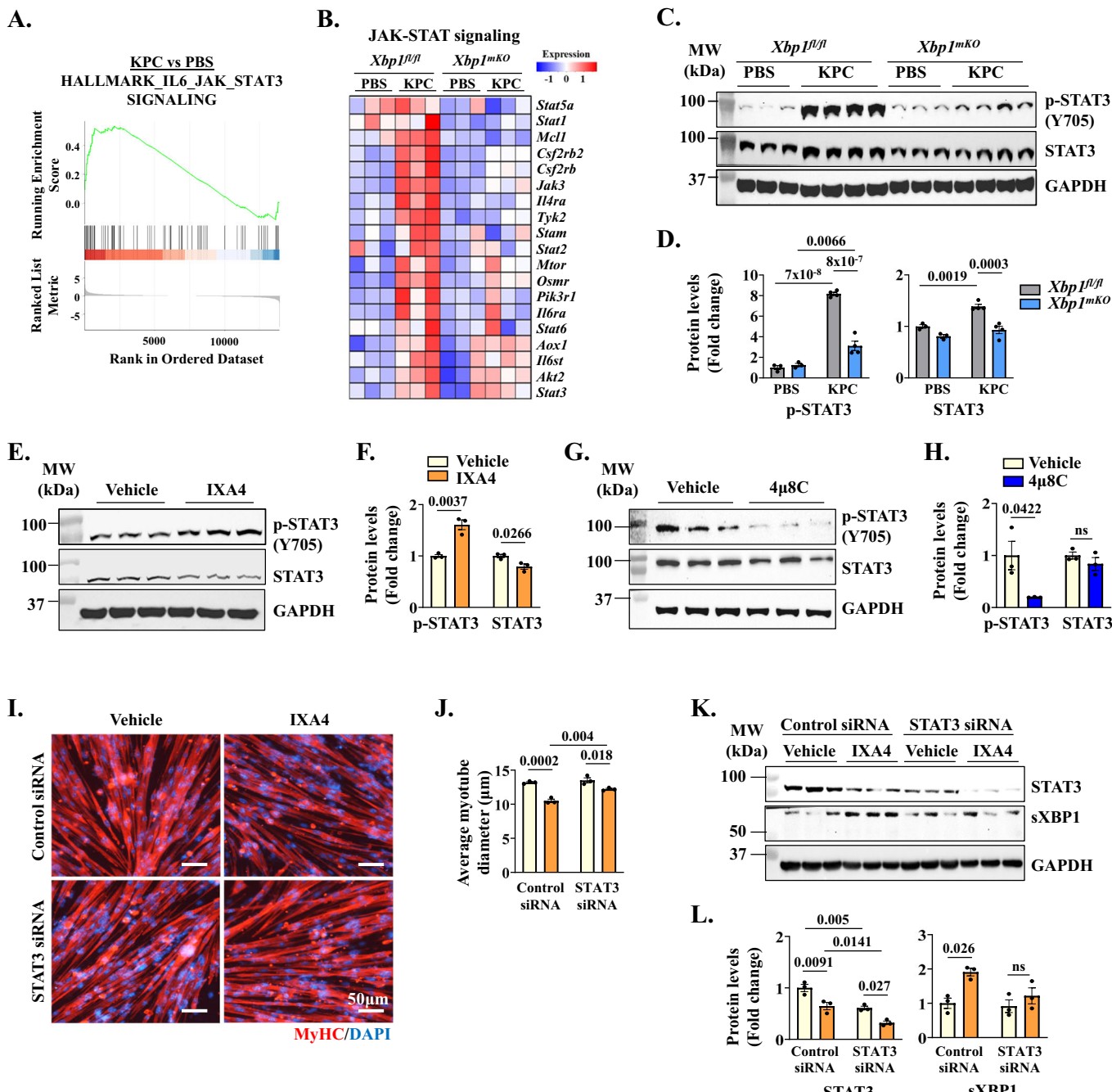

**Figure 5. Targeted deletion of XBP1 inhibits activation of IL6-JAK-STAT3 signaling in skeletal muscle during cachexia.**

(A) Gene Set Enrichment Analysis (GSEA) of DEGs shows upregulation in the gene set associated with IL6-JAK-STAT3 signaling in GA muscle of KPC tumor-bearing *Xbp1^fl/fl* mice compared to control *Xbp1^fl/fl* mice injected with PBS. (B) Heatmap representation shows gene expression of molecules involved in JAK-STAT signaling in GA muscle of control and KPC tumor-bearing *Xbp1^fl/fl* and *Xbp1^mKO* mice. Data information: Differentially expressed genes (DEGs) were identified using threshold of Log2FC ≥ 0.25 and *P* value < 0.05. (C) Immunoblots and (D) densitometry analysis showing levels of phosphorylated STAT3 (p-STAT3) and total STAT3 protein in GA muscle of control and KPC tumor-bearing *Xbp1^fl/fl* and *Xbp1^mKO* mice. *n* = 3–4 mice per group. Data information: Data are presented as mean ± SEM. Indicated *P* values were calculated using two-way ANOVA followed by Tukey's multiple comparison test. (E) Immunoblots and (F) densitometry analysis of protein levels of p-STAT3 and STAT3 in cultures treated with vehicle alone or 20 µM IXA4. *n* = 3 biological replicates per group. Data information: Data are presented as mean ± SEM. Indicated *P* values were calculated using unpaired Student *t* test. (G) Immunoblots and (H) densitometry analysis of protein levels of p-STAT3 and STAT3 in myotube cultures treated with vehicle alone or 4 µM 4µ8C. *n* = 3 biological replicates per group. Data information: Data are presented as mean ± SEM. Indicated *P* values were calculated using unpaired Student *t* test. (I) Representative photomicrographs of MyHC-stained myotube cultures. Scale bar, 50 µm, (J) quantification of myotube diameter in vehicle- or IXA4-treated cultures transfected with control or STAT3 siRNA, (K) immunoblots and (L) densitometry analysis showing protein levels of STAT3 and sXBP1 in vehicle- or IXA4-treated cultures transfected with control or STAT3 siRNA. *n* = 3 (biological replicates) per group. Data information: Data are presented as mean ± SEM. Indicated *P* values were calculated using two-way ANOVA followed by Tukey's multiple comparison test. Source data are available online for this figure.

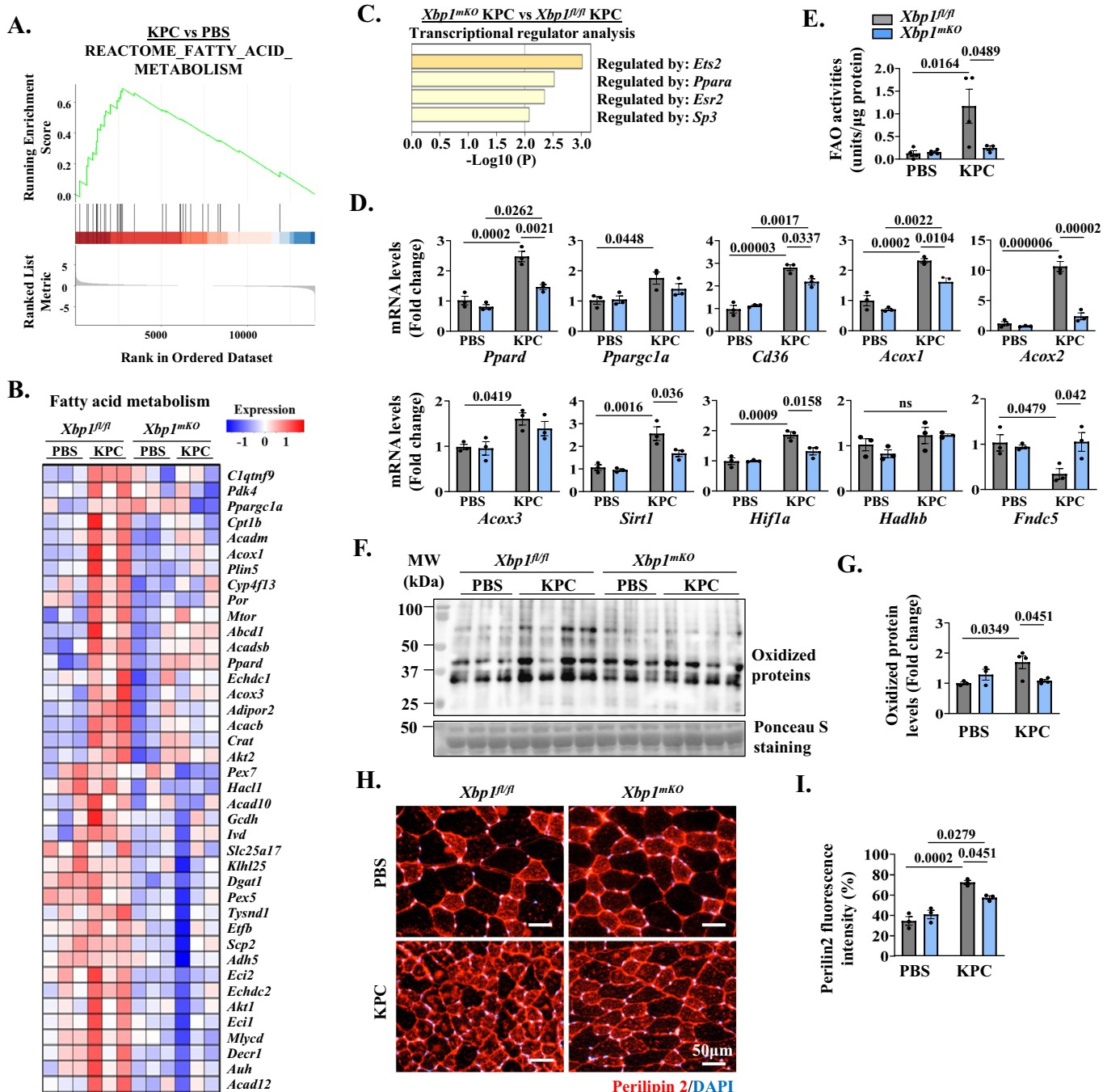

Figure A. KPC vs PBS — REACTOME_FATTY_ACID_METABOLISM

B. Fatty acid metabolism

C. *Xbp1^mKO* KPC vs *Xbp1^fl/fl* KPC — Transcriptional regulator analysis. Regulated by: *Ets2*; Regulated by: *Ppara*; Regulated by: *Esr2*; Regulated by: *Sp3*

D. mRNA levels (Fold change): *Ppard*, *Ppargc1a*, *Cd36*, *Acox1*, *Acox2*, *Acox3*, *Sirt1*, *Hif1a*, *Hadhb*, *Fndc5*

E. FAO activities (units/µg protein)

F. Oxidized proteins / Ponceau S staining

G. Oxidized protein levels (Fold change)

H. Perilipin 2/DAPI

I. Perilin2 fluorescence intensity (%)

*Xbp1^mKO* mice (Fig. 6D). Next, using a commercially available kit, we measured the fatty acid oxidation in skeletal muscle of control and KPC tumor-bearing *Xbp1^fl/fl* and *Xbp1^mKO* mice. Consistent with RNA-Seq results, there was a significant increase in the fatty acid oxidation in skeletal muscle of KPC tumor-bearing *Xbp1^fl/fl*, but not *Xbp1^mKO* mice. Importantly, fatty acid oxidation was significantly reduced in the skeletal muscle of tumor-bearing *Xbp1^mKO* mice compared with corresponding *Xbp1^fl/fl* mice (Fig. 6E).

Tumor-derived cachectic factors rapidly increase fatty acid metabolism which leads to oxidative stress and inhibition in muscle growth (Fang et al, 2023; Fukawa et al, 2016; Rohm et al, 2019). We

next investigated whether targeted ablation of XBP1 affects oxidative stress in skeletal muscle in response to KPC tumor growth. Interestingly, our results showed that there was a significant increase in the levels of carbonylated (irreversibly oxidized) protein in skeletal muscle of KPC tumor-bearing *Xbp1^fl/fl* mice compared to controls. Remarkably, the levels of carbonylated proteins were significantly reduced in the skeletal muscle of tumor bearing *Xbp1^mKO* mice compared with the corresponding *Xbp1^fl/fl* mice (Fig. 6F,G). Altogether, these results suggest that targeted ablation of XBP1 reduces fatty acid oxidation and oxidative stress in skeletal muscle of KPC tumor-bearing mice.

**Figure 6. Myofiber XBP1 induces fatty acid oxidation in cachectic muscle.**

(A) Gene Set Enrichment Analysis (GSEA) of DEGs shows upregulation in the gene set associated with fatty acid metabolism in GA muscle of KPC cells-injected *Xbp1^fl/fl* mice compared to PBS-injected *Xbp1^fl/fl* mice. (B) Heatmap showing relative mRNA levels of molecules involved in fatty acid oxidation in GA muscle of control and KPC tumor-bearing *Xbp1^fl/fl* and *Xbp1^mKO* mice. Data information: Differentially expressed genes (DEGs) were identified using threshold of Log2FC ≥ 0.25 and *P* value < 0.05. (C) Potential transcriptional regulators of downregulated genes in GA muscle of KPC tumor-bearing *Xbp1^mKO* mice compared to KPC tumor-bearing *Xbp1^fl/fl* mice identified using TRRUST database analysis. (D) Relative mRNA levels of PPARδ (gene name: *Ppard*), PGC1α (gene name: *Ppargc1a*), Cd36, Acox1, Acox2, Acox3, Sirt1, Hif1α, Hadhb, and Irisin (gene name: *Fndc5*) in GA muscle of control and KPC tumor-bearing *Xbp1^fl/fl* and *Xbp1^mKO* mice. n = 3 mice in each group. Data information: Data are presented as mean ± SEM. Indicated *P* values were calculated using two-way ANOVA followed by Tukey's multiple comparison test. (E) Quantification of fatty acid oxidation activity in skeletal muscle of control and KPC tumor-bearing *Xbp1^fl/fl* and *Xbp1^mKO* mice. n = 3–4 mice in each group. Data information: Data are presented as mean ± SEM. Indicated *P* values were calculated using two-way ANOVA followed by Tukey's multiple comparison test. (F) Immunoblot and (G) densitometry analysis of levels of carbonylated (irreversibly oxidized) protein in skeletal muscle of control and KPC tumor-bearing *Xbp1^fl/fl* and *Xbp1^mKO* mice. n = 3–4 mice in each group. Data information: Data are presented as mean ± SEM. Indicated *P* values were calculated using two-way ANOVA, followed by Tukey's multiple comparison test. (H) Representative photomicrographs and (I) quantification of anti-Perilipin 2 fluorescence signal in TA muscle sections of control and KPC tumor-bearing *Xbp1^fl/fl* and *Xbp1^mKO* mice after immunostaining for Perilipin 2 protein. n = 3–4 mice group. Data information: Data are presented as mean ± SEM. Indicated *P* values were calculated using two-way ANOVA followed by Tukey's multiple comparison test. Source data are available online for this figure.

Many recent studies have shown enhanced lipid accumulation into the skeletal muscle of animal models of cancer cachexia and human patients (Cardaci et al, 2024; Deng et al, 2024; Stephens et al, 2011). Interestingly, we found KPC tumor-induced increase in the gene expression of some of the molecules, especially Cd36, is significantly inhibited in *Xbp1^mKO* mice compared with *Xbp1^fl/fl* mice (Fig. 6D). To understand whether XBP1 has any role in regulating the lipid droplet content in skeletal muscle during cancer cachexia, we performed immunostaining for the Perilipin 2 protein, which is one of the most abundantly expressed lipid droplet-coating proteins in skeletal muscle (Bosma et al, 2012). Results showed that Perilipin 2 staining was significantly increased in TA muscle of KPC tumor-bearing mice compared to controls. Furthermore, we found that there was a small, but significant reduction in Perilipin 2 staining intensity on the TA muscle sections of KPC tumor-bearing *Xbp1^mKO* mice compared with corresponding *Xbp1^fl/fl* mice (Fig. 6H,I). These results suggest that in addition to fatty acid oxidation, the IRE1α/XBP1 axis regulates lipid content in skeletal muscle during pancreatic cancer cachexia.

## sXBP1 directly regulates the gene expression of specific molecules involved in muscle catabolism

The sXBP1 is a potent transcription factor which can directly interact with promoter region of various genes and regulate their expression (Hetz, 2012). Using UCSC genome browser, we first analyzed the presence of conserved sXBP1 binding sequences CCACG-box, UPR element (UPRE)-A, or UPRE-B motifs in the promoter region of genes whose expression is affected by the deletion of XBP1 in skeletal muscle of KPC tumor-bearing mice. Our analysis showed that the promoter region of muscle-specific E3-ubiquitin ligase MAFbx (*Fbxo32*) and autophagy-related genes (*Map1lc3b* and *Atg5*) contained sXBP1 binding domains that were conserved across different species. Interestingly, the promoter region of *Xbp1* gene itself has a CCACG motif (Fig. 7A,B). To determine whether sXBP1 can bind to the promoters of these genes during cachexia, we performed chromatin immunoprecipitation (ChIP) assay followed by semi-quantitative and quantitative PCR using primer sets designed to amplify 100–150 bp regions flanking the predicted binding sites. Interestingly, we observed significant enrichment of sXBP1 to the promoters of *Fbxo32*, *Map1lc3b*, *Atg5* and *Xbp1* genes in cultured myotubes following treatment with KPC-CM (Fig. 7C,D). The ChIP experiment was validated by

performing semi-quantitative PCR for positive control (*Rpl30* for Histone H3 antibody) and promoter sequences of known sXBP1 targets, *Dnajb9* and *Hspa5* (Fig. 7E).

We also analyzed the promoters of genes involved in inflammatory response and energy metabolism. We found sXBP1 conserved UPRE domains in the promoters of *Il6* and *Pdk4* whose products are involved in pro-inflammatory signaling (Eskiler et al, 2019; Wolsk et al, 2010), and glucose metabolism (Kim et al, 2023; Nahle et al, 2008), respectively (Fig. 7F). Since treatment of cultured myotubes with KPC-CM may not fully mimic the systemic inflammatory and metabolic alterations observed in vivo, we used GA muscles of KPC tumor-bearing mice to perform ChIP assay. Consistent with the in vitro results, we found enrichment of sXBP1 to the promoters of *Fbxo32* and *Xbp1* genes. Moreover, we observed sXBP1 enrichment in the promoter regions of *Il6*, and *Pdk4* (Fig. 7G,H). Even though the promoter region of *Stat3* gene also contains a conserved sXBP1 binding site, we did not find enrichment of sXBP1 to this site in the ChIP assay (Fig. 7F,G). Similar to cultured myotubes, we validated the ChIP assay with cachectic GA muscle using known target of sXBP1 (Fig. 7I). We further validated that forced activation of IRE1α/sXBP1 using IXA4 increases the mRNA levels of *Fbxo32*, *Map1lc3b*, *Atg5*, *Il6*, *Pdk4* and spliced-*Xbp* (*sXbp1*) in cultured myotubes (Fig. 7J). These results suggest that sXBP1 promotes expression of genes whose products are involved in cachectic muscle phenotype.

## Pharmacological inhibition of IRE1α/XBP1 axis attenuates KPC tumor-induced muscle wasting

We next sought to investigate whether pharmacological inhibition of IRE1α/XBP1 signaling can attenuate pancreatic cancer-induced muscle wasting in mice. Since tumor growth can affect the degree of cachexia, we first investigated whether IRE1α endonuclease activity inhibitor 4μ8C affects the proliferation of cultured KPC cells. Our results showed that treatment with 4μ8C did not alter the number of KPC cells till day 2 after treatment. However, there was a modest but significant reduction in the number of KPC cells by day 5 of 4μ8C treatment (Fig. EV5A) suggesting that inhibition of the IRE1α/XBP1 signaling can potentially inhibit KPC tumor growth in vivo. Using western blot analysis, we confirmed that the levels of sXBP1 protein are reduced in 4μ8C-treated KPC cultures (Fig. EV5B). To circumvent any inhibitory effect of 4μ8C on KPC tumorigenesis and growth in mice and to address the role of

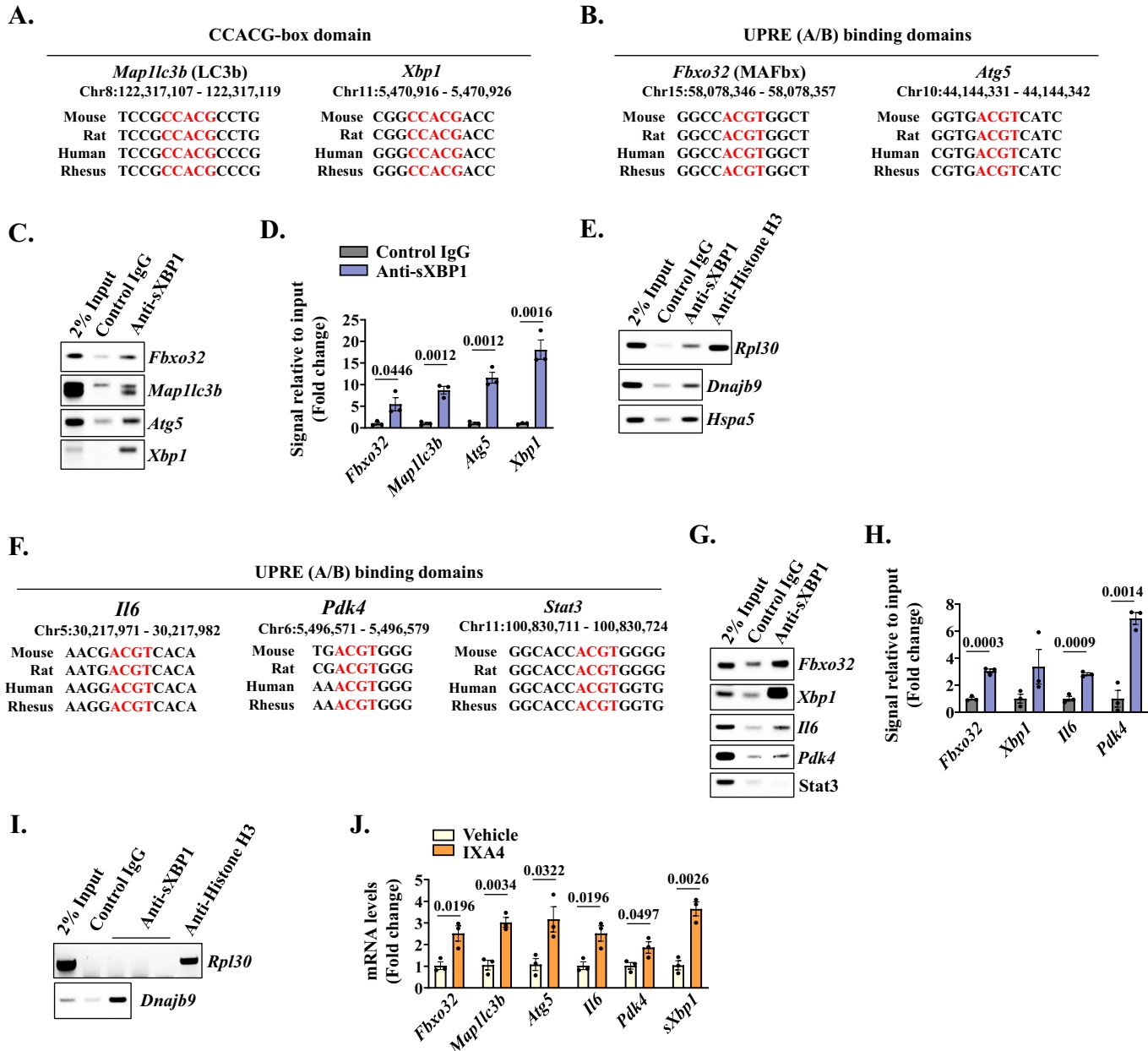

**Figure 7. Binding of sXBP1 to the promoters of specific genes whose products promote muscle wasting.**

(A, B) Conserved promoter regions containing sXBP1 binding sequences of (A) CCACG-box domains upstream of *Map1lc3b* (LC3B) and *Xbp1* genes, and (B) UPRE A/B domains upstream of *Fbxo32* (MAFbx) and *Atg5* genes are highlighted in red. (C, D) Chromatin immunoprecipitation (ChIP) showing enrichment of sXBP1 to the promoters of *Fbxo32*, *Map1lc3b*, *Atg5*, and *Xbp1* in KPC-CM-treated myotubes analyzed by performing (C) semi-quantitative PCR and (D) quantitative PCR (qPCR). $n = 3$ biological replicates per group. Data information: Data are presented as mean ± SEM. Indicated *P* values were calculated using unpaired Student *t* test. (E) Validation of ChIP assay in KPC-CM treated cultured myotubes was performed using PCR for Histone H3 antibody binding to the promoter of Rpl30 (positive control), and sXBP1 binding to promoter regions of known targets, *Dnajb9* and *Hspa5*. (F) Conserved promoter sequences containing sXBP1 UPRE A/B domains upstream of *Il6*, *Pdk4*, and *Stat3* genes, highlighted in red. (G) ChIP assay followed by PCR analysis shows enrichment of sXBP1 to the promoter regions of *Fbxo32*, *Xbp1*, *Il6*, *Pdk4*, and *Stat3* genes in GA muscle of KPC-tumor-bearing mice. (H) qPCR analysis showing fold increase in the enrichment of sXBP1 to the promoter regions of *Fbxo32*, *Xbp1*, *Il6*, *Pdk4* genes. GA muscle from 5 KPC tumor-bearing mice pooled for 1 biological sample. $n = 3$ biological replicates per group. Data information: Data are presented as mean ± SEM. Indicated *P* values were calculated using unpaired Student *t* test. (I) Validation of ChIP assay in GA muscle of tumor-bearing mice was performed using PCR for Histone H3 antibody binding to the promoter of *Rpl30* (positive control) and known sXBP1 target *Dnajb9*. (J) Relative mRNA levels of *Fbxo32*, *Map1lc3b*, *Atg5*, *Il6*, *Pdk4* and spliced-*Xbp1* in cultured myotubes after 8 h treatment with vehicle alone or 20 μM IXA4. $n = 3$ biological replicates per group. Data information: Data are presented as mean ± SEM. Indicated *P* values were calculated using unpaired Student *t* test. Source data are available online for this figure.

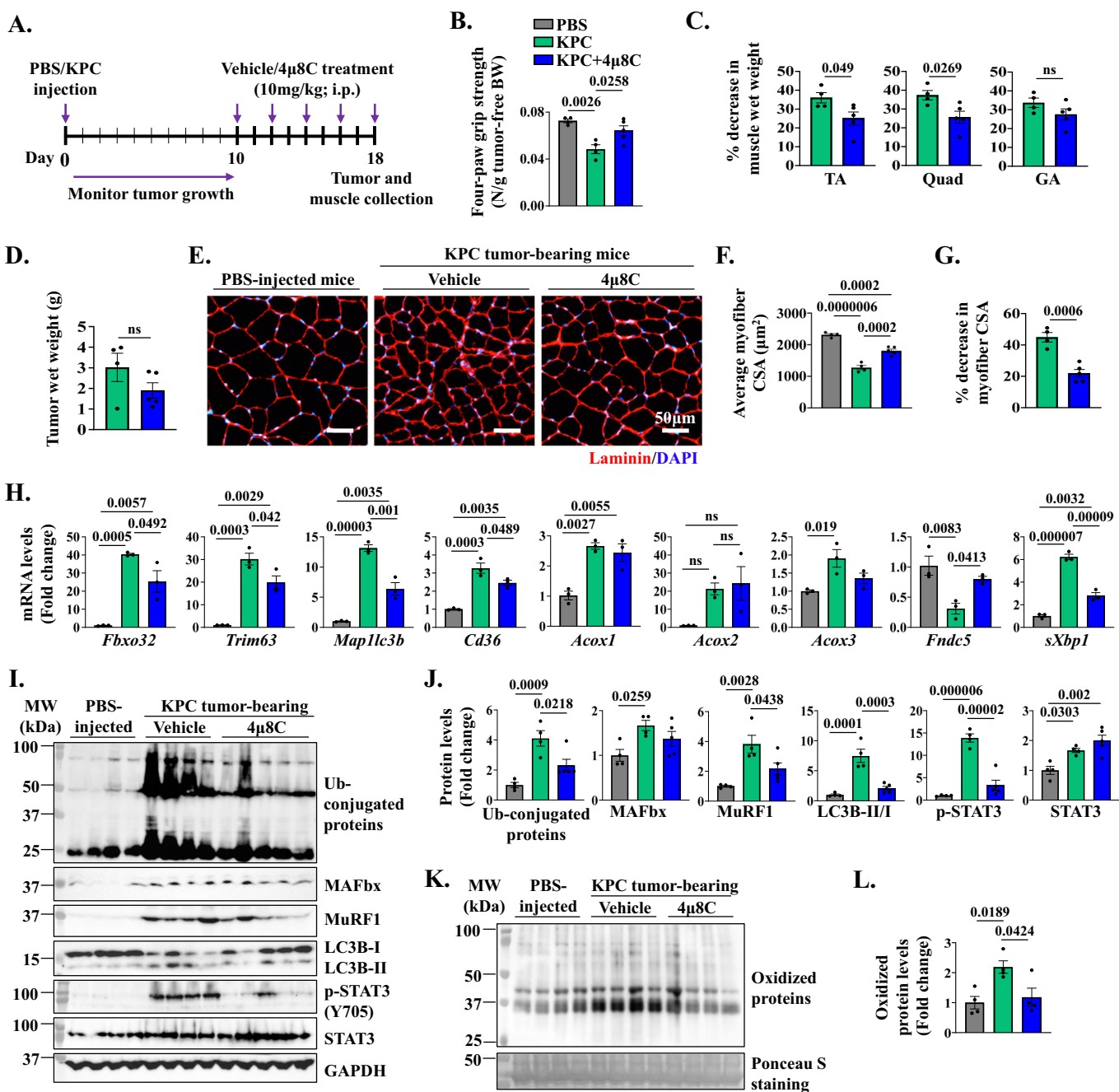

sXBP1 specifically in KPC tumor-induced muscle wasting, we started treatment of mice with 4μ8C when tumor was palpable (i.e., 10 days after injection of KPC cells). Briefly, wild-type mice were injected with PBS (control) or $2 \times 10^5$ KPC cells into the pancreas. On day 10, the tumor-bearing mice were randomly divided into two groups. One group of mice was intraperitoneally administered with 4μ8C drug (10 mg/kg body weight) every other day. The other group of tumor-bearing mice was administered with vehicle alone to serve as control (Fig. 8A). On day 18 after the injection of KPC cells in pancreas, the mice were analyzed for body weight and grip strength followed by collection of tumors and hindlimb muscles. Results showed that tumor-free body weight of both vehicle- and

4μ8C-treated KPC tumor-bearing mice was significantly reduced as compared to PBS-injected mice. However, there was no difference between the tumor-free body weight of vehicle- and 4μ8C-treated KPC tumor-bearing mice (Fig. EV5C). Interestingly, treatment with 4μ8C significantly improved four-paw grip strength in KPC tumor-bearing mice. Indeed, four paw grip strength normalized by tumor-free body weight of 4μ8C-treated KPC tumor-bearing mice was comparable to the levels of PBS-injected control mice (Fig. 8B).

Our analysis also showed that wet weight of TA, Quadriceps (Quad), and GA muscle normalized by tumor-free body weight, was significantly reduced in vehicle-treated KPC tumor bearing mice compared to control mice without tumor. Importantly,

**Figure 8. Pharmacological inhibition of IRE1α/XBP1 axis inhibits skeletal muscle wasting in KPC tumor-bearing mice.**

(A) Schematic representation of 4μ8C treatment regimen in KPC-tumor bearing mice. (B) Quantification of four-paw grip strength normalized by tumor-free body weight (BW). $n = 4$–5 mice in each group. Data information: Data are presented as mean ± SEM. Indicated *P* values were calculated using one-way ANOVA followed by Tukey's multiple comparison test. (C) Percentage decrease in wet weight of tibialis anterior (TA), quadriceps (Quad), and gastrocnemius (GA) muscle normalized by tumor-free BW of control (PBS-injected), and vehicle- or 4μ8C-treated KPC tumor-bearing mice. $n = 4$–5 mice in each group. Data information: Data are presented as mean ± SEM. Indicated *P* values were calculated using unpaired Student *t* test. (D) Quantification of tumor wet weight in KPC tumor-bearing mice treated with vehicle alone or 4μ8C. No significant difference was observed using the unpaired Student *t* test. (E) Transverse sections generated from TA muscle of control and vehicle- or 4μ8C-treated KPC tumor-bearing mice were immunostained for laminin protein. DAPI stain was used to identify nuclei. Representative photomicrographs are presented here. Scale bar, 50 μm. (F) Quantification of average myofiber CSA in TA muscle in vehicle- or 4μ8C-treated KPC tumor bearing mice compared with control mice. $n = 4$–5 mice in each group. Data information: Data are presented as mean ± SEM. Indicated *P* values were calculated using one-way ANOVA followed by Tukey's multiple comparison test. (G) Percentage decrease in average myofiber CSA in TA muscle in vehicle- or 4μ8C-treated KPC tumor bearing mice compared with control mice. $n = 4$–5 mice per group. Data information: Data are presented as mean ± SEM. Indicated *P* values were calculated using unpaired Student *t* test. (H) Relative mRNA levels of *Fbxo32*, *Trim63*, *Map1lc3b*, *Cd36*, *Acox1*, *Acox2*, *Acox3*, *Fndc5*, and spliced-*Xbp1* in skeletal muscle of control and vehicle- and 4μ8C-treated KPC tumor-bearing mice. $n = 3$ mice in each group. Data information: Data are presented as mean ± SEM. Indicated *P* values were calculated using one-way ANOVA followed by Tukey's multiple comparison test. (I) Immunoblots and (J) densitometry analysis of levels of ubiquitin-conjugated proteins, and MAFbx, MuRF1, LC3B-I/II, p-STAT3 and total STAT3 protein in TA muscle of control and vehicle- and 4μ8C-treated KPC tumor-bearing mice. $n = 4$–5 mice per group. Data information: Data are presented as mean ± SEM. Indicated *P* values were calculated using one-way ANOVA followed by Tukey's multiple comparison test. (K) Immunoblot, and (L) quantification of irreversibly oxidized proteins in TA muscle of control and vehicle- and 4μ8C-treated KPC tumor-bearing mice. $n = 4$ mice per group. Data information: Data are presented as mean ± SEM. Indicated *P* values were calculated by one-way ANOVA followed by Tukey's multiple comparison test. Source data are available online for this figure.

treatment with 4μ8C significantly inhibited the loss of wet weight of TA and Quad muscle in KPC tumor-bearing mice (Figs. 8C and EV5D). While 4μ8C treatment reduced the protein levels of sXBP1 in KPC tumor (Fig. EV5E,F), there was no significant difference in the average wet weight of KPC tumors in vehicle- and 4μ8C-treated mice (Fig. 8D). We also performed histological analysis of the tumor samples from vehicle- and 4μ8C-treated mice. There was no apparent difference in tumor histology between vehicle and 4μ8C groups analyzed by performing H&E staining (Fig. EV5G). Furthermore, there was no significant difference in the proportion of Ki67[+] nuclei in the tumor samples of vehicle- and 4μ8C-treated mice (Fig. EV5H,I) suggesting that 4μ8C treatment did not alter tumor growth in our model. To understand the effect of 4μ8C treatment on KPC tumor-induced muscle atrophy, we next generated transverse sections of TA muscle followed by performing H&E staining (Fig. EV5J) or immunostaining for laminin protein (Fig. 8E). Our analysis showed a drastic reduction in the average myofiber CSA of vehicle-injected KPC tumor-bearing mice compared to control mice. Importantly, treatment with 4μ8C significantly improved the average myofiber CSA in TA muscle of KPC tumor-bearing mice compared to corresponding mice treated with vehicle alone (Fig. 8F,G). We also measured mRNA levels of various components of UPS, autophagy, and fatty acid metabolism pathways in skeletal muscle of control and 4μ8C-treated mice. Results showed that KPC tumor-induced increase in the mRNA levels of *Fbxo32*, *Trim63*, *Map1lc3b*, *Cd36*, *Acox3*, and spliced-*Xbp1* (*sXbp1*) in skeletal muscle were significantly reduced by treatment of mice with 4μ8C. Moreover, 4μ8C treatment significantly increased the mRNA levels of Irisin in skeletal muscle of KPC tumor-bearing mice. In contrast, there was no significant difference in the mRNA levels of *Acox1* and *Acox2* between vehicle and 4μ8C-treated group (Fig. 8H). Western blot analysis showed that KPC tumor-induced increase in the levels of total ubiquitinated proteins, MAFbx, MuRF1, ratio of LC3B-II/I, and p-STAT3 protein in skeletal muscle were significantly reduced in 4μ8C-treated mice compared to corresponding controls treated with vehicle alone (Fig. 8I,J). Furthermore, treatment with 4μ8C significantly reduced the amounts of carbonylated proteins in skeletal of KPC tumor-bearing mice (Fig. 8K,L). Taken together, these results suggest that

pharmacological inhibition of sXBP1 ameliorates skeletal muscle wasting in the KPC mouse model of cancer cachexia.

# Discussion

Recent studies have shown that the markers of ER stress-induced UPR are elevated in the skeletal muscle of multiple animal models of cancer cachexia and in cancer patients (Roy and Kumar, 2019). Furthermore, it is now increasingly recognized that different arms of the UPR can play a distinct role in regulating skeletal muscle mass in various conditions (Afroze and Kumar, 2019). However, the role of the individual arms of the UPR in the regulation of skeletal muscle mass, especially during cancer-induced cachexia, remains poorly understood. We have previously reported that targeted inhibition of the PERK arm of the UPR does not affect the extent of muscle wasting in the LLC tumor-bearing mice (Gallot et al, 2019). While the mechanisms of action were not investigated, we showed that targeted deletion of XBP1 inhibits muscle wasting in response to LLC tumor growth (Bohnert et al, 2019). In the present study, we demonstrate that the IRE1α/XBP1 signaling axis is highly activated in skeletal muscle of the KPC mouse model of pancreatic cancer (Fig. 1). Our results using activators and inhibitors of IRE1α endonuclease activity suggest that IRE1α drives muscle wasting during pancreatic cancer cachexia mainly through promoting unconventional splicing of XBP1 mRNA which results in the production of transcriptionally active sXBP1 protein. Muscle-specific deletion of XBP1 is sufficient to improve skeletal muscle mass and strength in KPC tumor-bearing mice (Figs. 2 and 3). Moreover, our results demonstrate that pharmacological inhibition of the IRE1α/XBP1 arm of the UPR ameliorates the skeletal muscle wasting in response to KPC tumor growth (Fig. 8).

Muscle proteolysis in various conditions, including neoplastic growth, is attributed to increased activation of the UPS and autophagy-lysosomal system (Attaix et al, 1999; Penna et al, 2019; Penna et al, 2013; Zhang et al, 2020). While significant progress has been made towards understanding the signaling pathways involved in muscle wasting, molecular mechanisms that regulate the

expression of genes encoding various components of UPS and autophagy in skeletal muscle in catabolic conditions remain poorly understood. Our results in the present study demonstrate that XBP1 transcription factor regulates the gene expression of multiple components of the UPS and autophagy, including muscle-specific E3 ubiquitin ligases MAFbx and MuRF1, and LC3B. Muscle-specific deletion of XBP1 reduced the amounts of ubiquitinated proteins, levels of multiple components of the UPS and autophagy, strongly correlating with the preservation of muscle mass in KPC tumor-bearing mice (Fig. 4). Consistently, pharmacological inhibition of IRE1α/XBP1 axis using 4μ8C diminished the levels of ubiquitin-conjugated proteins, MAFbx and MuRF1, and LC3B (Fig. 8). These genetic and pharmacological approaches suggest that blockade of IRE1α/XBP1 axis preserves skeletal muscle mass by suppressing the activation of UPS and autophagy in KPC model of pancreatic cancer cachexia.

The IRE1α/XBP1 branch is the most conserved UPR pathway that resolves ER stress by increasing the folding capacity of the ER. However, in instances of high levels of unfolded or misfolded protein, XBP1 activates transcriptional programming of various components of the ERAD pathway (Park et al, 2021). ERAD, one of the important mechanisms to mitigate ER stress, targets unfolded and misfolded proteins through the ubiquitin–proteasome system and autophagy (Houck et al, 2014; Rashid et al, 2015; Wu and Rapoport, 2018). However, the role of ERAD pathway has not yet been studied in the context of muscle atrophy in catabolic conditions, including cancer. A recent study showed that the Sel1l-Hrd1 ERAD complex is required for muscle growth during postnatal development, suggesting its beneficial role in the regulation of muscle mass (Abdon et al, 2023). At present, it remains unknown whether the ERAD pathway is also involved in the etiology of cancer cachexia. Our RNA-seq analysis showed that the gene expression of multiple molecules involved in ERAD is upregulated in the skeletal muscle of KPC tumor-bearing mice and muscle-specific deletion of XBP1 dampens the expression of ERAD-related genes in response to KPC tumor growth (Fig. EV4C). Future studies will investigate the physiological significance of the ERAD in the regulation of skeletal muscle mass during cancer cachexia.

PDAC as well as other types of cancer and host cells secrete various molecules including proinflammatory cytokines that lead to local inflammation in various tissues, including skeletal muscle (Domaniku-Waraich et al, 2024; Liu et al, 2024; Rupert et al, 2021; Setiawan et al, 2023). IL-6 is one proinflammatory cytokine which induces skeletal muscle wasting in multiple models of cancer cachexia, including pancreatic cancer (Bonetto et al, 2012; Rupert et al, 2021). IL-6 activates the JAK-STAT3 signaling pathway that augments the activity of proteolytic systems in skeletal muscle (Bonetto et al, 2012; Eskiler et al, 2019; Silva et al, 2015). However, the mechanisms leading to increased activation of IL6-JAK-STAT signaling in cachectic muscle remain less understood. Intriguingly, our RNA-Seq analysis demonstrates that XBP1 regulates the gene expression of multiple components of this pathway, including cell surface receptors for IL-6 in cachectic muscle of KPC tumor-bearing mice (Fig. 5B). These results are also consistent with previously published reports demonstrating that XBP1 binds to the IL-6 promoter and activates its expression in melanoma and hepatocellular carcinoma (Chen and Zhang, 2017; Fang et al, 2018). The role of the IRE1α/XBP1 signaling in the activation of IL-6-

JAK-STAT signaling is also supported by the finding that myofiber-specific deletion of XBP1 strongly inhibited the phosphorylation of STAT3 in skeletal muscle of KPC tumor-bearing mice. Furthermore, IXA4, a pharmacological activator of IRE1α increased, whereas 4μ8C, an inhibitor of IRE1α endonuclease activity reduced the levels of phosphorylated STAT3 protein in cultured myotubes, suggesting that sXBP1 regulates the activation of STAT3 signaling both in vivo and in vitro. Our results also demonstrate that genetic knockdown of STAT3 abrogates IXA4-induced reduction in myotube diameter, further suggesting that IRE1α/XBP1 signaling mediates muscle wasting during pancreatic cancer cachexia through the activation of STAT3 signaling (Fig. 5).

Cancer cachexia also induces mitochondrial dysfunction and metabolic alterations in skeletal muscle (Martin et al, 2023; Setiawan et al, 2023). A prior study demonstrated that systemic inflammation during cachexia results in increased fatty acid oxidation in skeletal muscle of the C26 mouse model of cancer cachexia (Fukawa et al, 2016). Our results in the present study demonstrate that targeted ablation of XBP1 significantly reduced fatty acid oxidation and gene expression of various molecules involved in fatty acid metabolism in the skeletal muscle of KPC tumor-bearing mice. While fatty acid oxidation is an essential metabolic process for energy production, its excessive activation can lead to oxidative stress and muscle wasting during cancer cachexia (Ábrigo et al, 2018; Fukawa et al, 2016). Our results revealed that along with fatty acid oxidation, the levels of irreversibly oxidized proteins are increased in skeletal muscle of KPC tumor-bearing mice. Importantly, targeted ablation or pharmacological inhibition of XBP1 significantly reduced tumor-induced increase in oxidized proteins in skeletal muscle of mice (Figs. 6F,G and 8K,L). Since oxidative stress is an important stimulus for the activation of proteolytic systems and muscle wasting, it is possible that overstimulation of the IRE1α/XBP1 axis also causes muscle wasting through augmenting fatty acid oxidation and oxidative stress during cancer cachexia.

The sXBP1 protein is an active transcription factor that translocates to the nucleus and directly regulates the expression of its target genes by binding to the core UPRE-A, UPRE-B, or CCACG motif in the promoter regions (Acosta-Alvear et al, 2007; Park et al, 2021). ChIP-qPCR assay using myotubes treated with KPC-CM showed enrichment of sXBP1 to the conserved domains in the promoter regions of *Fbxo32*, *Map1lc3b*, and *Atg5* genes. A previous report showed that sXBP1 regulates autophagic responses in endothelial cells by binding to the promoter of Beclin1, an important component of autophagy. The study also showed that basal levels of LC3B are reduced in XBP1-deficient cells, however, no direct link between XBP1 and LC3B was reported (Margariti et al, 2013). Our results in this study demonstrate that sXBP1 binds to the promoter of *Map1lc3b*, which could be a potential mechanism to induce autophagy in skeletal muscle during cancer cachexia. Moreover, we also found enrichment of sXBP1 in the promoter region of MAFbx in skeletal muscle of KPC tumor-bearing mice. Intriguingly, our ChIP results, using both atrophying myotubes and cachectic mouse muscle, demonstrate that sXBP1 can bind to its own promoter suggesting a positive feedback loop to exacerbate muscle wasting (Fig. 7). Given that XBP1 pre-mRNA negatively regulates sXBP1 activity (Yoshida et al, 2006), it is also possible that sXBP1 induces XBP1 expression as an attempt to inhibit its hyperactivation during cancer cachexia.

While pharmacological inhibition of XBP1 is challenging, several small molecules targeting IRE1α endonuclease activity have been developed, including 4μ8C, which blocks the substrate access to the endoribonuclease domain of IRE1α, thereby inhibiting both splicing of XBP1 and RIDD-mediated mRNA degradation (Cross et al, 2012). We found that 4μ8C attenuates the loss of skeletal muscle mass and strength in the KPC mouse model of pancreatic cancer cachexia. Similar to muscle-specific *Xbp1*-knockout mice, treatment with 4μ8C diminishes the expression of various components of the UPS, autophagy, fatty acid metabolism, oxidative stress, and STAT3 phosphorylation in skeletal muscle of KPC tumor-bearing mice, further suggesting that sXBP1 promotes muscle wasting during cancer cachexia through the activation of proteolytic systems, fatty acid oxidation, and STAT3 signaling (Fig. 8).

The findings of the present study suggest a pivotal role of the IRE1α/XBP1 signaling in skeletal muscle wasting during cancer cachexia. Intriguingly, the IRE1α/XBP1 pathway also mediates myoblast fusion and regeneration of adult skeletal muscle (He et al, 2021; Joshi et al, 2024b; Roy et al, 2021). It is important to recognize that this pathway is only transiently activated during muscle regeneration in vivo or during differentiation of cultured myoblasts (Afroze and Kumar, 2019; Joshi et al, 2024b). Like many other signaling pathways, a transient increase in IRE1α-mediated signaling may be important for muscle regeneration and possibly for other physiological responses within skeletal muscle. It is the chronic activation or deregulation of such pathways that leads to deleterious effects, including muscle wasting in cancer-bearing hosts.

While our present study is focused on elucidating the role of IRE1α/XBP1 signaling in cachectic muscle, the UPR pathways are also known to regulate tumorigenesis, cancer progression, and resistance to chemotherapeutic drugs. Indeed, various components of ER stress-induced UPR pathways have been found to be highly upregulated in various types of cancers, including pancreatic cancer and a few studies have implicated the role of ER stress in the development of chemoresistance (Chen and Cubillos-Ruiz, 2021; Robinson et al, 2021). Increased lipid content has been observed in human pancreatic cancer and cell lines due to an increase in fatty acid synthase (FASN) expression which causes chemoresistance to gemcitabine, a widely used drug for the treatment of pancreatic cancer. The pharmacological inhibition of FASN significantly improves gemcitabine responsiveness mainly through the induction of ER stress that causes apoptosis (Tadros et al, 2017). Interestingly, pancreatic cancer cells can evade ER stress-induced apoptosis through upregulation of specific proteins, such as Mucin 1 and cytidine deaminase which are involved in pyrimidine metabolic pathway (Olou et al, 2020). However, most of the studies in pancreatic cancer have been performed using general ER stressors which activate all three arms of the UPR. It will be important to understand the role of individual arms of the UPR in the regulation of cancer cell survival. While our present study demonstrates that the inhibition of ER stress-induced IRE1α/XBP1 signaling attenuates muscle wasting during pancreatic cancer cachexia, the role of this pathway in the regulation of pancreatic cancer cell survival especially in response to chemotherapeutic agents needs further investigation.

In summary, our present study provides initial evidence that IRE1α/XBP1 signaling mediates skeletal muscle wasting during

pancreatic cancer-induced cachexia. Our results also corroborate that IRE1α/XBP1 axis regulates multiple mechanisms that have a causative role in skeletal muscle wasting. Future studies will determine whether similar mechanisms are involved in muscle wasting in other models of cancer cachexia and pancreatic cancer patients.

# Methods

### Reagents and tools table

| Reagent/resource | Reference or source | Identifier or catalog number |
| --- | --- | --- |
| **Experimental models** | | |
| C57BL/6 J mice (Mus musculus) | Jackson Lab | RRID:IMSR_JAX:000664 |
| MCK-Cre Mice/ Tg(Ckmm-cre)5Khn | Jackson Lab | RRID:IMSR_JAX:006475 |
| *Xbp1*fl/fl mice/ *Xbp1*tm2Glm | Dana-Farber Cancer Institute, Boston, MA | RRID:MGI:3774017 |
| Primary mouse myoblast cultures | This study | N/A |
| KPC cell line | Obtained from Elizabeth Jaffee (Johns Hopkins University, Baltimore, MD) | N/A |
| **Antibodies** | | |
| Rabbit anti-MAFbx (WB, 1:1000) | ECM Biosciences | Cat # AP2041 |
| Goat anti-MuRF1 (WB, 1:1000) | R&D Systems | Cat # AF5366 |
| Rabbit anti-Beclin1 (WB, 1:1000) | Cell Signaling | Cat # 3495S |
| Rabbit anti-LC3b (WB, 1:1000) | Cell Signaling | Cat # 2775S |
| Mouse anti-Ubiquitin (WB, 1:1000) | Santa Cruz | Cat # sc-8017 |
| Rabbit anti-IRE1alpha (WB, 1:1000) | Cell Signaling | Cat # 3294S |
| Rabbit anti-sXBP1 (E9V3E) (WB, 1:1000) | Cell Signaling | Cat # 40435 |
| Rabbit anti-PERK (WB, 1:1000) | Cell Signaling | Cat # 3192 |
| Mouse anti-CHOP (WB, 1:1000) | Cell Signaling | Cat # 2895S |
| Rabbit anti-p-STAT3 (Y705) (WB, 1:1000) | Cell Signaling | Cat # 9145 |
| Rabbit anti-STAT3 (WB, 1:1000) | Cell Signaling | Cat # 30835 |
| Rabbit anti-GAPDH (WB, 1:1000) | Cell Signaling | Cat # 2118 |
| Anti-rabbit IgG HRP (WB, 1:2000) | Cell Signaling | Cat # 7074S |
| Anti-mouse IgG HRP (WB, 1:2000) | Cell Signaling | Cat # 7076S |
| Anti-goat IgG HRP (WB, 1:2000) | Invitrogen | Cat # A15999 |
| Rabbit anti-Laminin (IF, 1:1000) | Sigma Chemical Co. | Cat # L9393 |

| Reagent/resource | Reference or source | Identifier or catalog number |
|---|---|---|
| Mouse anti-Myosin heavy chain (IF, 1:150) | DSHB | Cat # MF20 |
| Rabbit anti-Perilipin-2 (IF, 1:300) | Proteintech | Cat # 15294-1-AP |
| Anti-Mouse IgG2b AF555 (IF, 1:1500) | Invitrogen | Cat # A21147 |
| Anti-rabbit IgG AF555 (IF, 1:1500) | Invitrogen | Cat # A31572 |
| **Oligonucleotides and other sequence-based reagents** | | |
| XBP1 siRNA (m) | Santa Cruz Biotechnology Inc. | Cat # sc-38628 |
| STAT3 siRNA (m) | Santa Cruz Biotechnology Inc. | Cat # sc-29494 |
| Mouse Xbp1 (exon 2) Forward Primer-5'-CCTGAGCCCGGAGGAGAA-3' | Chen et al, 2018 | N/A |
| Mouse Xbp1 (exon 2) Reverse Primer-5'-CTCGAGCAGTCTGCGCTG -3' | Chen et al, 2018 | N/A |
| Other PCR primers | This study | Table EV1 |
| **Chemicals, enzymes and other reagents** | | |
| Colchicine | Cayman Chemical | Cat # 9000760 |
| Chloroquine diphosphate | Tocris Bioscience | Cat # 4109 |
| IXA4 | MedChemExpress | Cat # HY-139214 |
| 4μ8C | MedChemExpress | Cat # HY-19707 |
| Lipofectamine™ RNAimax Transfection Reagent | Invitrogen | Cat # 13778150 |
| OxyBlot Protein Oxidation Detection Kit | Millipore Sigma | Cat # S7150 |
| Fatty Acid Oxidation (FAO) Assay Kit | AssayGenie | Cat # BR00001 |
| Picro-Sirius Red Stain Kit (For Collagen) | StatLab | Cat # KTPSRPT |
| **Software** | | |
| Cell Ranger v3.1.0 | 10X Genomics | https://support.10xgenomics.com/single-cell-gene-expression/software/overview/welcome |
| Seurat v3.1.1 | Butler et al, 2018 | https://satijalab.org/seurat/ |
| Metascape | Zhou et al, 2019 | https://metascape.org/ |
| **Other** | | |
| Bulk RNA-seq (Deposited data) | This study | SRA: PRJNA1255232 |

## Mice

C57BL/6 wild-type (WT) mice were purchased from Jackson Laboratories (Bar Harbor, ME, USA). Floxed Xbp1 (Xbp1^fl/fl) mice were crossed with MCK-Cre (Strain: B6.FVB (129S4)-Tg (Ckm-cre) 5Khn/J, Jackson Laboratory, Bar Harbor, ME) mice to generate muscle-specific XBP1 knockout (Xbp1^mKO) and littermate control (Xbp1^fl/fl mice) as described (Bohnert et al, 2019; Roy et al, 2021). All mice were in the C57BL/6 background, and their genotype was determined by PCR from tail DNA. Mice were housed in individual

cages in an environmentally controlled room (23 °C, 12-h light-dark cycle) with ad libitum access to food and water. We employed KPC orthotopic mouse model of pancreatic cancer cachexia (Michaelis et al, 2017). Briefly, KPC cells ($2 \times 10^5$ cells in 20 μl PBS) were injected into the tail of the pancreas of 10–12-week-old WT or Xbp1^fl/fl and Xbp1^mKO mice. Control mice received an injection of PBS alone. The mice were weighed weekly and euthanized on day 18–21 after the injection of KPC cells. For pharmacological studies, 10 days after injection of KPC cells, the mice were given i.p. injections of 4μ8C (diluted in 5% DMSO and 95% Corn Oil; 10 mg/kg) or vehicle alone (control) every other day. All the animals were handled according to the approved institutional animal care and use committee (IACUC) protocol (PROTO201900043) of the University of Houston. All surgeries were performed under anesthesia, and every effort was made to minimize suffering.

## Grip strength test

Four-paw grip strength of mice was measured used a digital grip strength meter (Columbus Instruments, Columbus, OH), as described (Sato et al, 2015). Briefly, mice were acclimatized for 5 min before the grip strength test. The mouse was allowed to grab the metal pull bar with all 4 paws. The tail of the mouse was then gently pulled backward in the horizontal plane until it could no longer grasp the bar. The force at the time of release was recorded as the peak tension. Each mouse was tested 5 times with a 30-second break between tests. The average peak tension from the 5 repetitions was normalized against body weight.

## Autophagy flux assay

In vivo assessment of autophagic flux was performed using colchicine treatment as described (Ju et al, 2010). Briefly, 4 mg/mL stock solution of colchicine was prepared in water and stored at −20 °C. On the day of treatment, the stock was diluted to 0.1 mg/mL in water for injection. Control and KPC tumor-bearing mice were given i.p. injections of 0.4 mg/kg/day colchicine 24 h and 4 h prior to euthanizing the mice and collecting muscle tissues. Control mice received an equal volume of water via i.p. injections.

## Measurement of protein synthesis

The mice were anesthetized and given i.p. injection of 0.04 μm of puromycin per gram of body weight. The mice were euthanized exactly 30 min after injection of puromycin, and hind limb muscle was isolated and snap frozen. For cultured myotubes, 1 μM puromycin was added in the culture medium for 30 min. Finally, protein extracts were made from muscles or cultured myotubes, and newly synthesized protein was detected by performing immunoblotting using primary antibody anti-puromycin.

## Histology and morphometric analysis

Individual TA and soleus muscles were isolated from mice, snap-frozen in liquid nitrogen, and sectioned with a microtome cryostat. For the assessment of muscle morphology, 8-μm-thick transverse sections of TA and soleus muscle were stained with hematoxylin and eosin (H&E) dye. To assess collagen deposition in muscle tissues, Sirius red

staining was performed using a commercially available kit (StatLab, McKinney, Texas, USA). The sections were examined under Nikon Eclipse Ti-2E Inverted Microscope (Nikon). Muscle sections were also processed for immunostaining for laminin protein to mark the boundaries of myofibers. For immunostained sections, the slides were mounted using fluorescence medium (Vector Laboratories) and visualized at room temperature on Nikon Eclipse Ti-2E Inverted Microscope (Nikon), a digital camera (Digital Sight DS-Fi3, Nikon), and Nikon NIS Elements AR software (Nikon). Average myofiber cross-sectional area (CSA) was analyzed in anti-laminin-stained muscle sections using ImageJ software (NIH, Bethesda, MD). For each muscle, average CSA was calculated by analyzing ~200 myofibers measured in each muscle per mouse, and frequency distribution curve was plotted for categorical intervals of 200 $\mu m^2$ myofiber CSA. H&E staining was also performed on tumor cryosections to visualize gross histology of tumors. For quantification of proliferating cells in tumors, anti-Ki67 and DAPI staining was performed and the regions of highest density Ki67+ cells (i.e., hotspots) were selected for enumerating Ki67+ cells per unit area ($mm^2$).

## Intramuscular lipid droplet staining

To assess the abundance of intramuscular lipid droplets, Perilipin 2 immunostaining was performed on TA muscle transverse sections. Briefly, the muscle sections were fixed in 4% PFA in PBS for 15 min, followed by rinsing with PBS and blocking with 3% BSA solution. Subsequently, muscle sections were incubated with Perilipin 2 (Plin2) antibody (Proteintech, Rosemont, IL) overnight at 4 °C in a humidified chamber. The Perilipin 2 staining was detected using AF555 anti-rabbit IgG. Nuclei were counterstained with DAPI. The Plin2 fluorescence intensity was quantified using ImageJ software (NIH, Bethesda, MD).

## Measurement of fatty acid oxidation (FAO)

FAO activity in skeletal muscle was measured using fatty acid oxidation (FAO) assay kit (Assay Genie, Dublin, Ireland) following manufacturer's suggested protocol.

## Carbonylation assay

The carbonyl groups in the protein side chains were derivatized to 2,4-dinitrophenylhydrazone by reaction with 2,4-dinitrophenylhydrazine using OxyBlot Protein Oxidation Detection Kit (Sigma-Aldrich, St. Louis, Missouri, USA). The carbonylated/oxidatively modified proteins were detected by performing immunoblotting using anti–2,4-dinitrophenyl hydrazone antibodies.

## KPC cultures

KPC cells, as described (Foley et al, 2015), were kindly provided by Elizabeth Jaffee (Johns Hopkins University, Baltimore, MD) and cultured in RPMI1640 supplemented with 10% FBS. For preparation of KPC cells conditioned medium (KPC-CM), the cells were cultured to confluency, incubated in differentiation medium (DM, DMEM with 2% horse serum) for 24 h, followed by collection of media in the supernatants. The media was clarified of suspended cells by centrifugation and passing through a sterile 0.22 μm syringe filter. The KPC-CM was diluted 1:4 in fresh DM for inducing

atrophy in cultured myotubes. For in vitro assessment of autophagy flux, cultured myotubes were treated with 100 μM chloroquine for 2 h prior to collection, as described earlier (Gallot et al, 2019).

## Myotube cultures and Immunostaining analysis

Primary myoblasts were isolated from hindlimb muscle of C57BL/6 mice as described (Hindi et al, 2017a). For myotube formation, primary myoblasts were incubated in DM for 48 h. To assess protein synthesis, puromycin incorporation in myotubes was performed by adding puromycin (1 μg/mL) to culture medium for 30 min followed by collection of cells for immunoblotting. Myotubes were transfected with control, IRE1α, XBP1, or STAT3 siRNA (Santa Cruz Biotechnology) using Lipofectamine RNAiMAX Transfection Reagent (Thermo-Fisher Scientific) for 24 h. For pharmacological treatments, myotube cultures were treated with 4μ8C or IXA4. For immunostaining, the cultures were fixed with 4% PFA in PBS for 15 min at room temperature and permeabilized with 0.1% Triton X-100 in PBS for 10 min. Cells were blocked with 2% bovine serum albumin in PBS and incubated with mouse-anti-MyHC (clone MF20, DSHB, Iowa City, Iowa) overnight at 4 °C. The cells were then washed with PBS and incubated with a secondary antibody at room temperature for 1 h. Nuclei were counterstained with DAPI for 3 min. The stained cultures were photographed and analyzed using Nikon Eclipse Ti-2E Inverted Microscope (Nikon), a digital camera (Digital Sight DS-Fi1), and Elements BR 3.00 software (Nikon). Image levels were equally adjusted using Photoshop CS6 software (Adobe). Average myotube diameter was calculated by measuring diameter of 100-120 MyHC+-myotubes per group using the NIH ImageJ software. For consistency, diameters were measured at the midpoint along with the length of the MyHC+ myotubes.

## Western blot

GA muscle of mice or cultured primary myotubes were washed with PBS and homogenized in lysis buffer (50 mM Tris-Cl (pH 8.0), 200 mM NaCl, 50 mM NaF, 1 mM dithiothreitol, 1 mM sodium orthovanadate, 0.3% IGEPAL, and protease inhibitors). Approximately, 100 μg protein was resolved on each lane on 10–12% SDS-PAGE gel, transferred onto a nitrocellulose membrane, and probed using a specific primary antibody. Bound antibodies were detected by secondary antibodies conjugated to horseradish peroxidase (Cell Signaling Technology). Signal detection was performed by an enhanced chemiluminescence detection reagent (Bio-Rad). Approximate molecular masses were determined by comparison with the migration of prestained protein standards (Bio-Rad).

## RNA extraction and qRT-PCR

RNA isolation and qRT-PCR were performed following a standard protocol as described (Hindi et al, 2017b). In brief, total RNA was extracted from GA muscle of control or KPC tumor-bearing $Xbp1^{fl/fl}$ and $Xbp1^{mKO}$ mice or cultured myotubes using TRIzol reagent (ThermoFisher Scientific) and RNeasy Mini Kit (Qiagen, Valencia, CA, USA) according to the manufacturer's protocols. First-strand cDNA for PCR analysis was made with a commercially available kit (iScript cDNA Synthesis Kit, Bio-Rad Laboratories). The quantification of mRNA expression was performed using the SYBR Green dye (Bio-Rad SsoAdvanced-Universal SYBR Green

**The paper explained**

**Problem**

Although the loss of skeletal muscle mass is a major contributor to morbidity and mortality in cancer patients, the molecular mechanisms driving cancer-induced muscle wasting remain less understood.

**Results**

Pancreatic cancer progression activates the IRE1/XBP1 signaling pathway in skeletal muscle. Both genetic and pharmacological inhibition of this pathway significantly reduce muscle wasting in a preclinical model of pancreatic cancer.

**Impact**

Targeting the IRE1/XBP1 signaling axis represents a promising therapeutic strategy to counteract cancer-associated muscle wasting and potentially improve outcomes for patients with pancreatic cancer.

Supermix) method on a sequence detection system (CFX384 Touch Real-Time PCR Detection System, Bio-Rad Laboratories). The sequence of the primers is described in Table EV1. Data normalization was accomplished with the endogenous control (β-actin), and the normalized values were subjected to a 2-ΔΔCt formula to calculate the fold change between control and experimental groups.

## ChIP assay

ChIP was performed using SimpleChIP enzymatic Chromatin IP kit (Cell Signaling Technology, Cat #9003) according to the manufacturer's suggested protocol. Briefly, primary myotube cultures were treated KPC-CM for 24 h followed by crosslinking, purification of nuclei, and chromatin shearing by sonication (eight times, 20 s each). For processing cachectic muscle isolated from KPC tumor-bearing mice, GA muscle from five mice were pooled, minced in hypotonic buffer, homogenized using loose Dounce homogenizer (25-30 strokes), followed by crosslinking, purification of nuclei, and chromatin shearing. Three independent sets of mice were used as biological replicates. The following steps remained the same for cultured cells or tissue samples. Sheared chromatin was incubated overnight with antibodies against sXBP1 (Cell Signaling Technology) or Histone H3 (Cell Signaling Technology) followed by incubation with Protein G magnetic beads. Normal Rabbit IgG (Cell Signaling Technology) was used as a negative control for the immunoprecipitation experiment. Magnetic beads were briefly washed, and chromatin DNA was eluted, reverse crosslinked, and purified. Purified DNA was analyzed for enrichment of 100–150 bp sequences by quantitative real time-PCR (40 cycles) or semi-quantitative standard PCR using specific primers designed for binding sites in the regulatory regions of Fbxo32, Map1lc3b, Atg5, Xbp1, Il6, Pdk4, Hspa5, Dnajb9, and mouse Rpl30 genes. The fold change between negative control IgG and anti-sXBP1 groups was calculated using $2 - \Delta\Delta Ct$ formula normalized by signal from 2% input. Primers used for PCR reactions are provided in supplemental Table EV1.

## RNA sequencing and data analysis

Total RNA from GA muscle of PBS- or KPC-injected $Xbp1^{fl/fl}$ and $Xbp1^{mKO}$ mice was extracted using TRIzol reagent (ThermoFisher

Scientific) using the RNeasy Mini Kit (Qiagen, Valencia, CA, USA) according to the manufacturer's protocols. The mRNA-seq library was prepared using poly (A)-tailed enriched mRNA at the UT Cancer Genomics Center using the KAPA mRNA HyperPrep Kit protocol (KK8581, Roche, Holding AG, Switzerland) and KAPA Unique Dual-indexed Adapter kit (KK8727, Roche). The Illumina NextSeq550 was used to produce 75 base paired-end mRNA-seq data at an average read depth of ~38 M reads/sample. RNA-seq fastq data were processed using CLC Genomics Workbench 20 (Qiagen). Illumina sequencing adapters were trimmed, and reads were aligned to the mouse reference genome Refseq GRCm39.105 from the Biomedical Genomics Analysis Plugin 20.0.1 (Qiagen). Normalization of RNA-seq data were performed using a trimmed mean of M values. Genes with $\text{Log}_2\text{FC} \geq |0.25|$ and $P$ value $< 0.05$ were assigned as differentially expressed genes (DEGs) and represented in volcano plot using ggplot function in R software (v 4.2.2). Pathway enrichment analysis was performed using the Metascape Gene Annotation and Analysis tool (metascape.org) as described (Zhou et al, 2019). Heatmaps were generated by using heatmap.2 function (Gu and Hübschmann, 2022) using z-scores calculated based on transcripts per million (TPM) values. The average absolute TPM values for the control group are provided in Table EV2. TPM values were converted to log (TPM + 1) to handle zero values. Genes involved in specific pathways were manually selected for heatmap expression plots. All the raw data files can be found on the NCBI SRA repository using the accession code PRJNA1255232.

## Statistical analysis and experimental design

The sample size was calculated using power analysis methods for a priori determination based on the SEM and the effect size was previously obtained using the experimental procedures employed in the study. For animal studies, we calculated the minimal sample size for each group as eight animals. Considering the likely drop-off effect of 10%, we set the sample size of each group of six mice. For some experiments, three to four animals were sufficient to obtain statistically significant differences. Animals of the same sex and same age were employed to minimize physiological variability and to reduce s.d. from the mean. The exclusion criteria for animals were established in consultation with a veterinarian and experimental outcomes. In case of death, skin injury, ulceration, sickness, or weight loss of >10%, the animal was excluded from the analysis. Tissue samples were excluded in cases such as freezing artifacts on histological sections or failure in the extraction of RNA or protein of suitable quality and quantity. We included animals from different breeding cages by random allocation to the different experimental groups. All animal experiments were blinded using number codes until the final data analyses were performed. Statistical tests were used as described in the Figure legends. Results are expressed as mean ± SEM. Statistical analyses used unpaired Student's $t$ test, or one-way or two-way ANOVA followed by Tukey's multiple comparison test to compare quantitative data populations with normal distribution and equal variance. The value of $P < 0.05$ was considered statistically significant unless otherwise specified.

## Data availability

RNA-seq data generated for this study have been deposited in the SRA database with the accession number PRJNA1255232.

The source data of this paper are collected in the following database record: biostudies:S-SCDT-10_1038-S44321-025-00337-w.

## Peer review information

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

## Acknowledgements

We sincerely thank Dr. Laurie Glimcher of Dana-Farber Cancer Institute for providing floxed Xbp1 mice. This work was supported by the National Institute of Health grant AR081487 and CA294365 to AK.

## Author contributions

**Aniket S Joshi**: Data curation; Formal analysis; Investigation; Methodology; Writing—original draft. **Meiricris Tomaz da Silva**: Data curation; Validation; Investigation; Methodology; Writing—review and editing. **Anh Tuan Vuong**: Data curation; Investigation; Methodology. **Bowen Xu**: Data curation; Investigation; Methodology. **Ravi K Singh**: Conceptualization; Supervision; Writing—review and editing. **Ashok Kumar**: Conceptualization; Formal analysis; Supervision; Funding acquisition; Project administration; Writing—review and editing.

Source data underlying figure panels in this paper may have individual authorship assigned. Where available, figure panel/source data authorship is listed in the following database record: biostudies:S-SCDT-10_1038-S44321-025-00337-w.

## Disclosure and competing interests statement

The authors declare no competing interests.

# Expanded View Figures

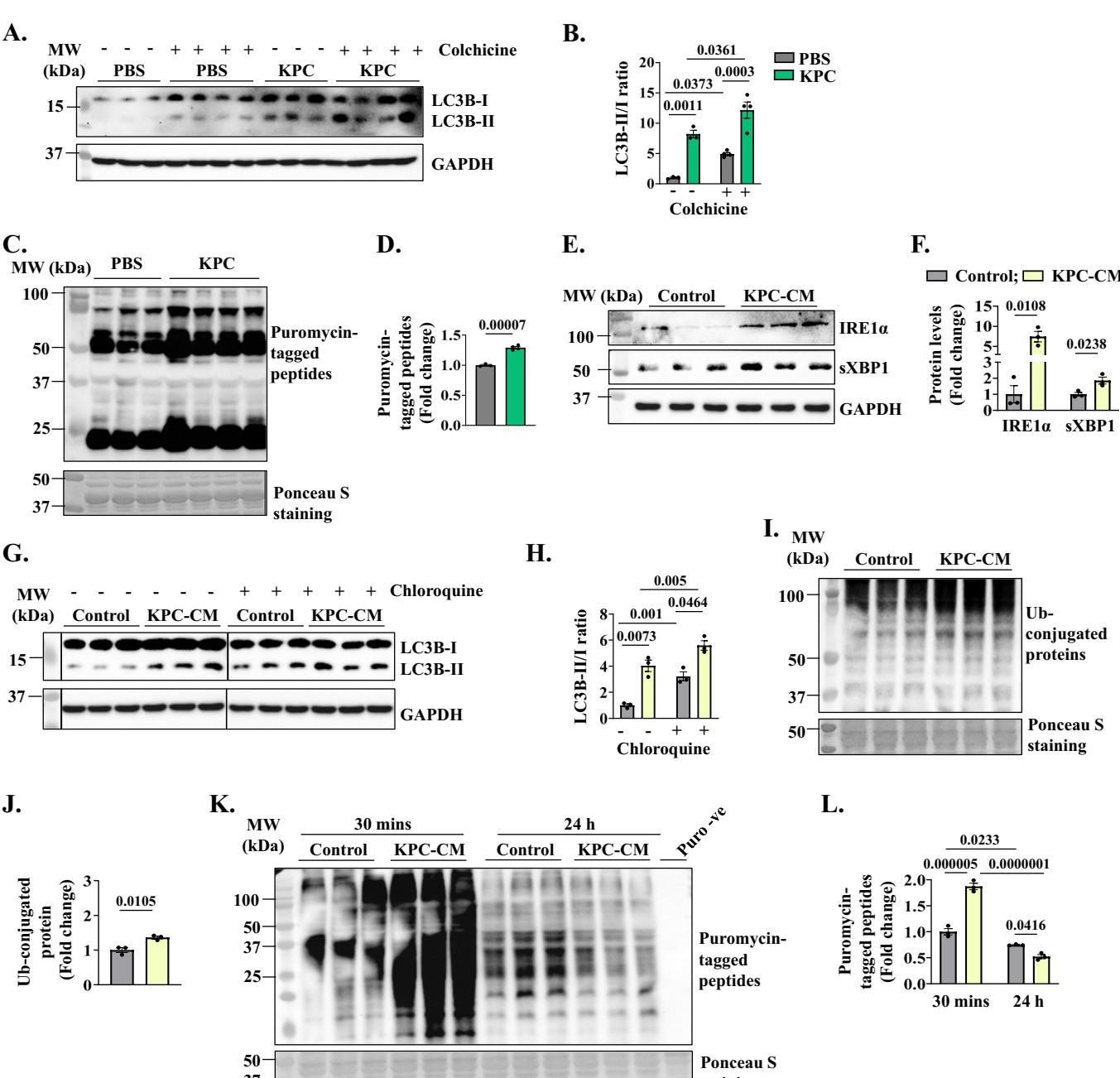

**Figure EV1. KPC tumor-derived factors activate IRE1α/XBP1 signaling and regulate protein turnover in cultured myotubes.**

(A) Immunoblots and (B) densitometry analysis of levels of LC3B protein in control (PBS) and KPC tumor-bearing mice with or without treatment with (0.4 mg/kg/day) colchicine. $n = 3$–4 mice in each group. Data information: Data are presented as mean ± SEM. Indicated $P$ values were calculated using two-way ANOVA followed by Tukey's multiple comparison test. (C) Immunoblots and (D) densitometry analysis of amounts of puromycin-tagged peptides in control (PBS) and KPC tumor-bearing mice. $n = 3$–4 mice in each group. Data information: Data are presented as mean ± SEM. Indicated $P$ values were calculated using unpaired Student $t$ test. (E) Immunoblots and (F) densitometry analysis of levels of IRE1α and sXBP1 protein in control and KPC-CM treated myotube cultures. (G) Immunoblot and (H) quantification of ratio of LC3B-II/I protein in myotube cultures incubated in DM (control) or KPC-CM and treated with vehicle alone or (100 μM) chloroquine. $n = 3$ biological replicates per group. Data information: Data are presented as mean ± SEM. Indicated $P$ values were calculated using two-way ANOVA followed by Tukey's multiple comparison test. (I) Immunoblots and (J) densitometry analysis of levels of ubiquitin (Ub)-conjugated proteins in control and KPC-CM treated cultures. $n = 3$ biological replicates per group. Data information: Data are presented as mean ± SEM. Indicated $P$ values were calculated using unpaired Student $t$ test. (K) Immunoblots and (L) densitometry analysis of amounts of puromycin-tagged peptides in myotube cultures after 30 mins or 24 h of incubation in DM (control) or KPC-CM. $n = 3$ biological replicates in each group. Data information: Data are presented as mean ± SEM. Indicated $P$ values were calculated using two-way ANOVA followed by Tukey's multiple comparison test. Source data are available online for this figure.

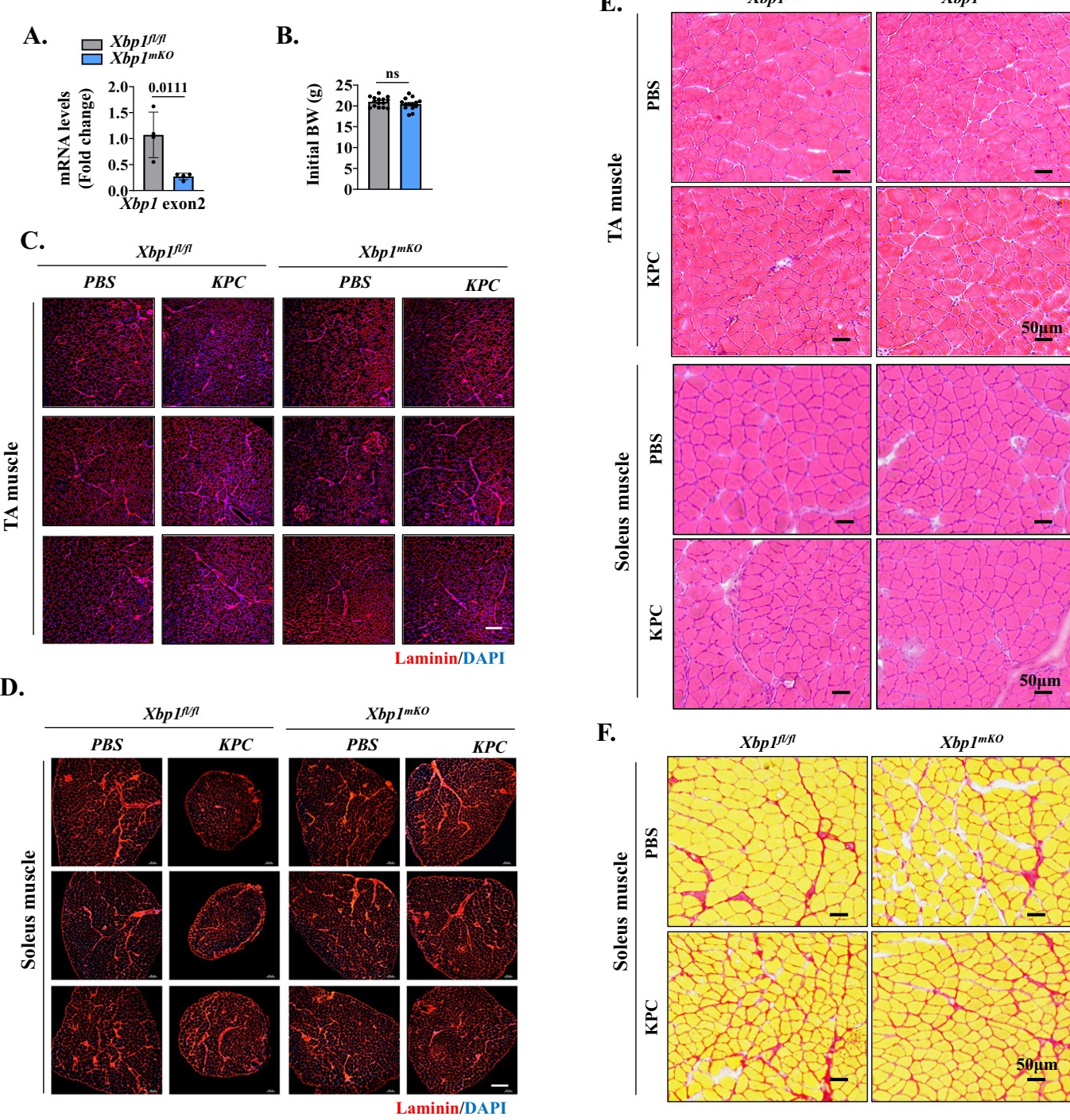

**Figure EV2.  Targeted deletion of XBP1 inhibits muscle atrophy in KPC tumor-bearing mice.**

(A) Relative mRNA levels of XBP1 determined by qRT-PCR analysis using primer set specific for Xbp1 exon 2, a sequence flanked by the loxP sites, in gastrocnemius (GA) muscle of $Xbp1^{fl/fl}$ and $Xbp1^{mKO}$ mice. $n = 4$ mice per group. Data information: Data are presented as mean ± SEM. Indicated $P$ values were calculated using unpaired Student $t$ test. (B) Quantification of initial body weight of $Xbp1^{fl/fl}$ and $Xbp1^{mKO}$ mice. $n = 14$–15 mice per group. Data information: No significant difference was observed using unpaired Student $t$ test. Transverse sections of TA and soleus muscle isolated from PBS- or KPC cells-injected $Xbp1^{fl/fl}$ and $Xbp1^{mKO}$ mice were used for anti-laminin and DAPI staining or H&E staining. (C, D) Anti-laminin and DAPI stained sections of (C) TA and (D) soleus muscle from multiple mice. Scale bar, 200μm. (E) Representative photomicrographs of H&E-stained transverse sections of TA (upper panel) and soleus (lower panel) muscle of control and KPC tumor-bearing $Xbp1^{fl/fl}$ and $Xbp1^{mKO}$ mice. Scale bar, 50μm. (F) Representative photomicrographs of Sirius red-stained soleus muscle sections of $Xbp1^{fl/fl}$ and $Xbp1^{mKO}$ mice injected with PBS or KPC cells. Scale bar, 50 μm. Source data are available online for this figure.

**A.** Ubiquitin proteasome system

**B.** Autophagy/Mitophagy

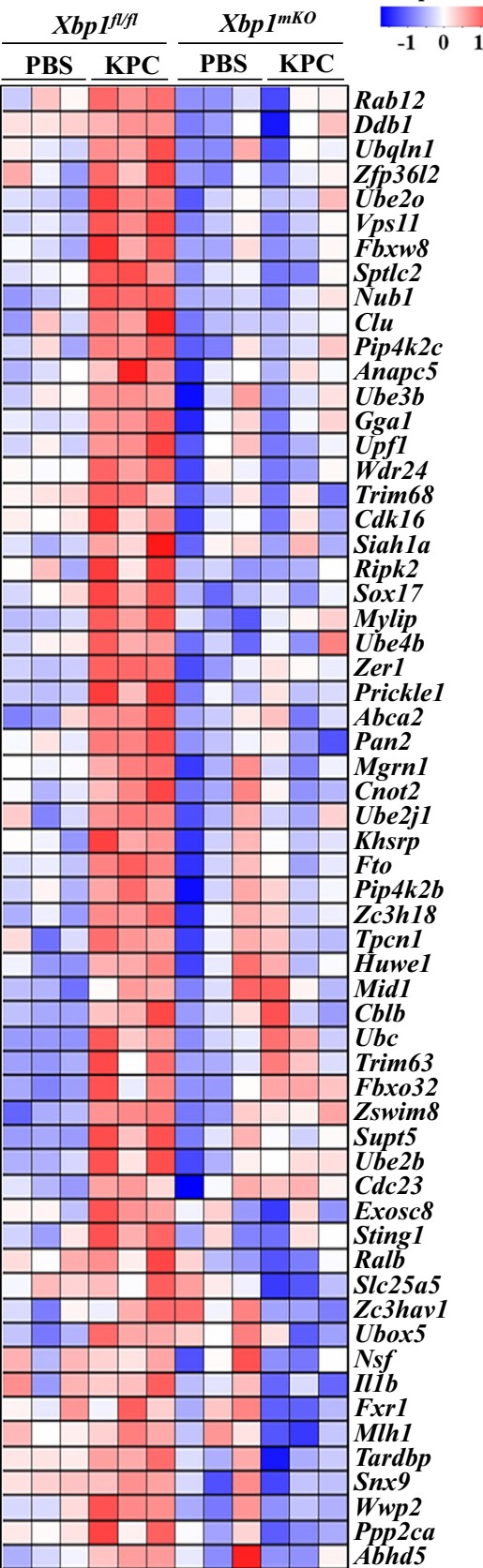

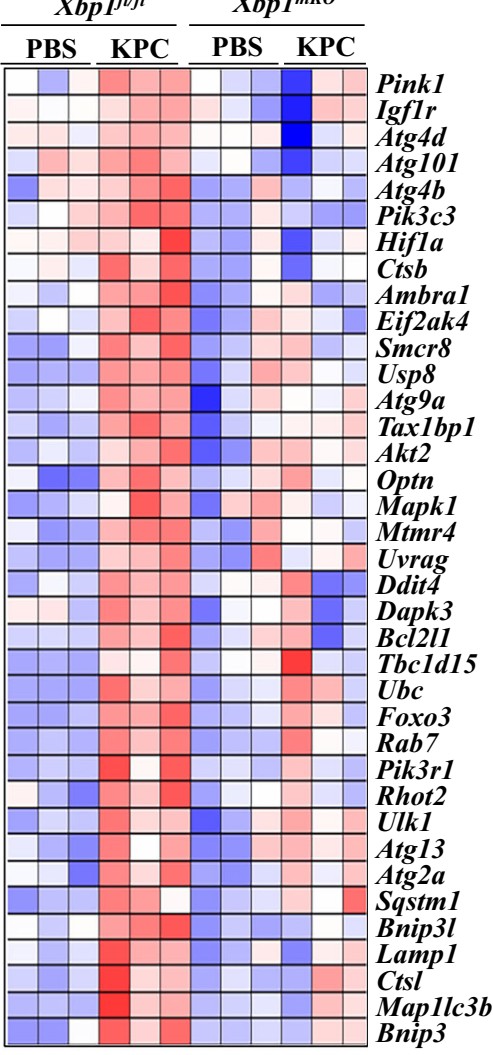

◄ **Figure EV3. Targeted ablation of XBP1 inhibits the expression of multiple genes involved in proteolysis.**

(A, B) Heatmap representation of RNA-Seq dataset analysis showing relative expression of genes involved in (A) Ubiquitin proteasome system (UPS), and (B) autophagy/mitophagy in gastrocnemius muscle of control and KPC tumor-bearing *Xbp1*<sup>fl/fl</sup> and *Xbp1*<sup>mKO</sup> mice.

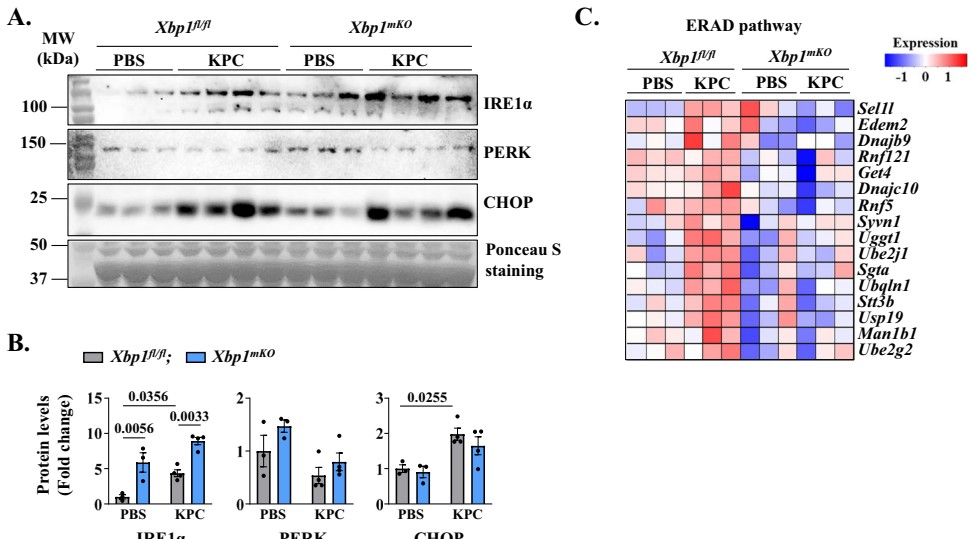

**Figure EV4.  Effect of targeted ablation of XBP1 on levels of UPR markers.**

(A) Immunoblots and (B) densitometry analysis of protein levels of IRE1α, PERK, and CHOP in gastrocnemius (GA) muscle of *Xbp1*^fl/fl^ and *Xbp1*^mKO^ mice injected with PBS or KPC cells. *n* = 3–4 mice per group. Data information: Data are presented as mean ± SEM. Indicated *P* values were calculated using two-way ANOVA followed by Tukey's multiple comparison test. (C) Heatmap representing relative gene expression of various molecules involved in ER associated degradation (ERAD) pathway in GA muscle of control and KPC tumor-bearing *Xbp1*^fl/fl^ and *Xbp1*^mKO^ mice analyzed using bulk RNA-seq dataset. Source data are available online for this figure.

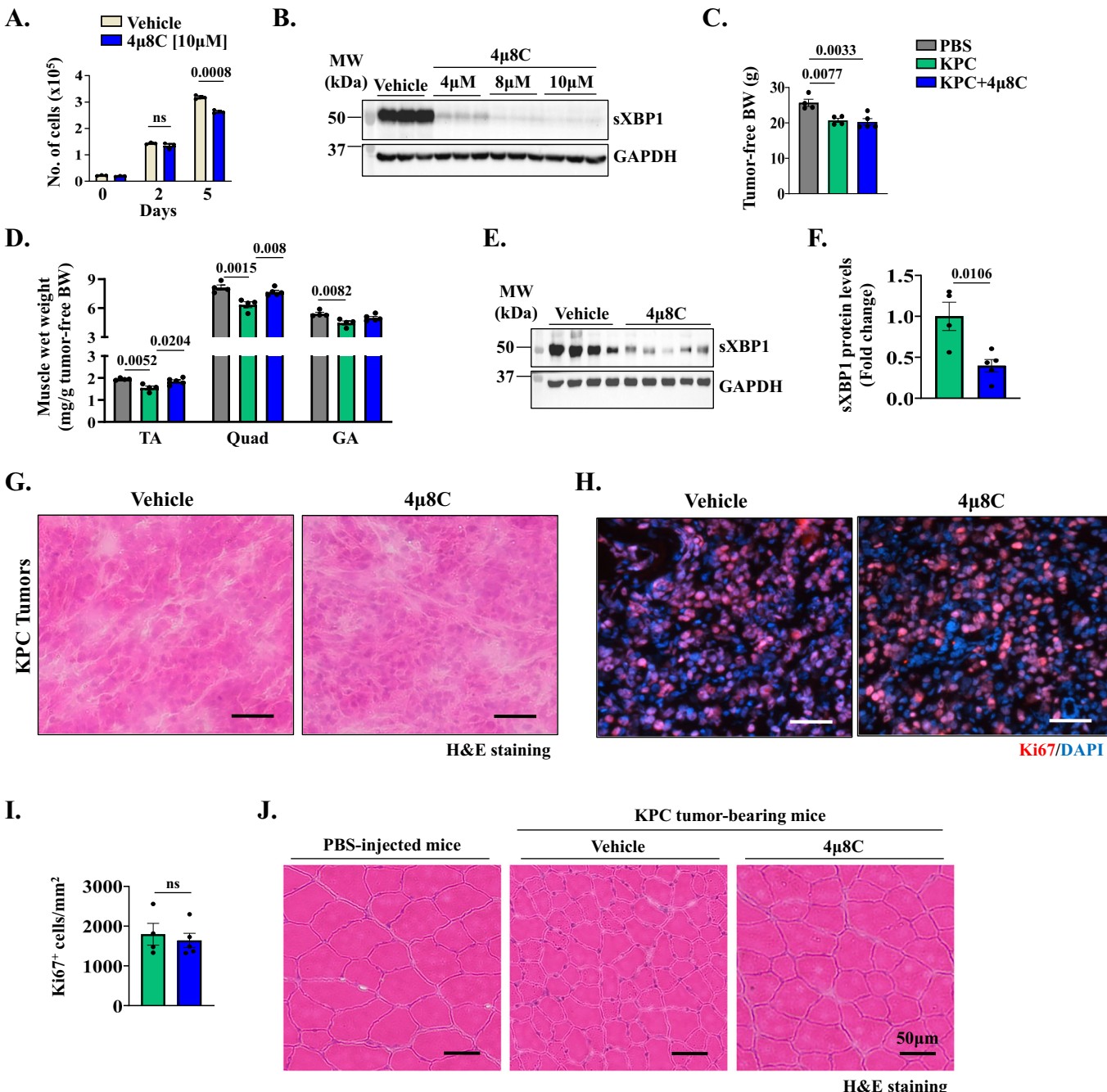

**Figure EV5. Effect of pharmacological inhibition of IRE1α/XBP1 axis on cancer cachexia.**

(A) Quantification of number of proliferating KPC cells on day 0, 2 and 5 after treatment with vehicle alone or 10 μM 4μ8C. $n = 3$ biological replicates per group. Data information: Data are presented as mean ± SEM. Indicated *P* values were calculated using unpaired Student *t* test. (B) Immunoblot showing levels of sXBP1 protein in KPC cells treated with vehicle alone or indicated concentrations of 4μ8C for 24 h. (C) Quantification of tumor-free body weight (BW) of control and KPC tumor-bearing mice treated with vehicle alone or 4μ8C after 18 days of KPC cells injection into the pancreas. (D) Quantification of wet weight of TA, Quad, and GA muscle normalized by tumor-free BW. $n = 4$–5 mice per group. Data information: Data are presented as mean ± SEM. Indicated *P* values were calculated using one-way ANOVA, followed by Tukey's multiple comparison test. (E) Immunoblot and (F) densitometry analysis showing levels of sXBP1 protein in KPC tumors of mice treated with vehicle alone or 4μ8C. Data information: Data are presented as mean ± SEM. Indicated *P* values were calculated using unpaired Student *t* test. (G, H) Representative photomicrographs of KPC tumors after (G) H&E staining, or (H) anti-Ki67 and DAPI staining. Scale bar, 50μm. (I) Quantification of number of Ki67[+] cells per unit area (mm²) in KPC tumors of mice treated with vehicle alone of 4μ8C. Data information: No significant difference was observed using unpaired Student *t* test. (J) Transverse sections of TA muscle of control and vehicle or 4μ8C-treated KPC tumor-bearing mice were generated and used for H&E staining. Representative photomicrographs of H&E-stained sections are presented here. Scale bar, 50 μm. Source data are available online for this figure.

