## [Peer Review File · EMBO Molecular Medicine]

The canonical ER stress IRE1 α /XBP1 pathway mediates skeletal muscle wasting during pancreatic cancer cachexia

Aniket Joshi, Meiricris Tomaz da Silva, Anh Vuong, Bowen Xu, Ravi Singh, and Ashok Kumar

Corresponding author: Ashok Kumar (akumar43@central.uh.edu)

Review Timeline:

Submission Date:	25th May 25
Editorial Decision:	20th Jun 25
Revision Received:	4th Aug 25
Editorial Decision:	20th Aug 25
Revision Received:	22nd Sep 25
Editorial Decision:	17th Oct 25
Revision Received:	30th Oct 25
Accepted:	31st Oct 25

Editor: *Zeljko Durdevic*

Transaction Report:

20th Jun 2025

Dear Prof. Kumar,

Thank you for the submission of your manuscript to EMBO Molecular Medicine. We have now received feedback from two of the three reviewers who agreed to evaluate your manuscript. As the referee #2 despite several reminders will unfortunately not be able to return his/her report in a timely manner, we prefer to make a decision now in order to avoid further delay in the process.

As you will see from their reports pasted below, both referees recognize potential interest of the study but also raise serious concerns. Given the nature of the referees' criticism addressing all the referees' comments, particularly functional validation of RNA Seq data, would require a lot of additional work, time, and effort. Therefore, I am afraid that we do not feel it would be productive to call for a revised version of your manuscript.

That being said and given the potential interest of the findings, we would, however, be willing to consider a new manuscript on the same topic if at some time in the near future you obtained data that would considerably strengthen the message of the study and address the referees concerns in full. To be completely clear, however, I would like to stress that if you were to send a new manuscript this would be treated as a new submission rather than a revision and would be reviewed afresh, in particular with respect to the literature and the novelty of your findings at the time of resubmission. If you decide to follow this route, please make sure you nevertheless upload a letter of response to the referees' comments.

I am sorry that I could not bring better news this time and hope that the referee comments are helpful in your continued work in this area.

Yours sincerely,

Zeljko Durdevic

Zeljko Durdevic
Senior Editor
EMBO Molecular Medicine

**** Reviewer's comments ****

Referee #1 (Comments on Novelty/Model System for Author):

The transgenic mice being utilized are optimal for these studies.

Referee #1 (Remarks for Author):

Targeting the canonical ER stress IRE1 α /XBP1 pathway counteracts pancreatic cancer-induced skeletal muscle wasting

This study has elucidated a novel pathway to address ER stress-associated muscle wasting in pancreatic cancer. Interestingly, this study demonstrates that the IRE1/XBP1 pathway is highly activated in the skeletal muscle of KPC tumor-bearing mice. Interestingly, this study identifies XBP1 as a key transcription factor that directly binds to the promoters of multiple cachexia-related genes, as well as its own promoter, in the KPC model, suggesting a positive feedback loop. Mechanistically, sXBP1 (spliced XBP1) enhances expression of genes involved in the ubiquitin-proteasome system (UPS) and autophagy, such as MAFbx, MuRF1, and LC3b. ChIP-qPCR revealed that sXBP1 binds to the promoter regions of these genes, thereby strengthening the hypothesis of this manuscript and indicating transcriptional regulation. Deletion or inhibition of XBP1 also suppressed components of the ER-associated degradation (ERAD) pathway and reduced activation of the proinflammatory IL-6-JAK-STAT3 signaling axis, further linking ER stress to cachectic signaling. Treatment with 4 μ 8C, an IRE1 endonuclease inhibitor, recapitulated the protective effects seen with Xbp1 knockout, attenuating proteolysis, autophagy, fatty acid oxidation, and STAT3 signaling. These findings position the IRE1/XBP1 axis as a central regulator of skeletal muscle wasting and a promising therapeutic target in PDAC-associated cachexia. Overall, the manuscript presents novel findings with strong supporting data, providing detailed mechanistic insights that implicate the IRE1/XBP1 pathway in cachexia. Appropriate controls have been included, and the manuscript is well-written, presenting novel findings with robust scientific rigor. Utilization of multiple transgenic mouse models is another strength. I have the following comments:

Major comments:

1. Figure 1 presents initial findings on ER stress using the gastrocnemius muscle. It would strengthen Figure 2 to include Laminin or H&E staining and cross-sectional area (CSA) measurements of the gastrocnemius muscle as well.
2. In Figure 2C, there is no change in muscle weight of TA and Soleus between Xbp1^{fl/fl} and Xbp1^{mKO}, but a significant decrease in myofiber CSA (Figure 2E-J), which is a marker of muscle mass loss. The authors should explain this discrepancy.
3. In Figure 2A, there is no change in tumor-free body weight between Xbp1^{fl/fl} and Xbp1^{mKO}; body weight loss is a primary phenotype of cancer cachexia. Is this due to it being an early time point? Please include an explanation in the discussion.
4. In Figure 3 J and L, why is there a difference in the protein expression of sXBP1 in the vehicle groups? Perhaps somewhat comparable exposures for vehicle controls would be useful.
5. Including tumor volume data in the study would enhance its comprehensiveness.
6. Figure 8I, blot quality is not good specifically p-STAT3. I would recommend changing these blots with clear ones.
7. Previous studies have demonstrated that inducing ER stress can be beneficial in eradicating PDAC tumors by chemotherapy, and tumor cells have mechanisms to overcome ER stress (PMID: 28811332, 32103170). The authors should briefly describe the tumor-centric impact of the inhibitor and cite these articles in the context of systemic ER stress inhibition.

Minor comments:

1. The text conversion to PDF created some erroneous squares in the text in the introduction, line 7 on page 3, the last sentence on page 4, page 24, and page 25. Please correct.
2. Reference 17 states intramyocellular lipid droplet accumulation in cachexia. This does not fit with the sentence "Indeed, depletion of intramyocellular lipids, due to accelerated fatty acid metabolism, is a characteristic feature of the cachectic muscles of animal models of cancer cachexia (16-18)." Please change the description or the reference.
3. I would recommend mentioning the physiological role/status of IRE1 α and XBP1 in the introduction.
4. In the abstract, the sentence "Transcriptionally active XBP1 protein binds to the promoter region of multiple genes whose products are involved in skeletal muscle wasting", would be clearer if key target genes name were specified.

Referee #3 (Remarks for Author):

The present study by Joshi and collaborators is aimed to investigate the relevance of the IRE1 α /XBP1 axis to the onset and progression of cachexia induced by an implantable model of pancreatic cancer (KPC). The Authors previously showed that ER stress contributes to cancer-induced cachexia (PMID: 27206451) and that the IRE1 α /XBP1 axis is overactivated in the muscle of mice bearing the Lewis Lung Carcinoma (PMID: 31138662). The study by Joshi et al. extends the focus to cachexia induced by the KPC tumor. Most of the conclusions refer to results obtained by the RNAseq analysis performed on the gastrocnemius muscle. The hypothesis that the IRE1 α /XBP1 axis could play a causative role in KPC-induced cachexia is supported, partially at least, by experiments in which the axis is modulated by means of genetic or pharmacological tools.

Points to clarify/integrate/improve:

- the data showing that markers of protein turnover are up-regulated at both mRNA and protein levels just suggest that the system is altered. A dynamic assessment of protein turnover rates should be provided; even if not totally explanatory, measuring the autophagic flux and protein synthesis by puromycin incorporation will give an acceptable estimate;
- the Authors show that the IRE1 α /XBP1 axis is able to regulate STAT3 signaling in the muscle of tumor-bearing mice. The anatomical compartment (the muscle) is the only novel issue in this regard, since IRE1 α /XBP1-dependent activation of STAT3 was previously demonstrated in cancer cells (PMID: 28222747; PMID: 30214606);
- page 11, the Authors state: The GSEA of RNA-seq dataset showed that the fatty acid metabolism was significantly increased. This sentence is totally misleading, since RNAseq data do not demonstrate a metabolic activity. This example very well fits with the overall impression reported above, that many conclusions are drawn on the basis of mRNA expression data, while little insight into biological activity is provided;
- surprisingly, the muscles of KPC hosts (Xbp1^{fl/fl}) show a low intramyocellular lipid content (Oil-Red-O positive lipid droplets). Such a pattern is reverted when the KPC tumor is implanted into Xbp1^{mKO}. This is somewhat in contrast with other data in the literature (PMID: 39010842; PMID: 21766057), also in cancer patients (PMID: 38725139; PMID: 36189019; PMID: 35220502, among others);
- page 17, the Authors define ERAD as another proteolytic pathway, which is not totally correct. Indeed, proteasome and autophagy are the main actors involved in ERAD. By contrast, ERAD is activated by peculiar mechanisms, which can make the difference with respect to the 'canonical' FoxO/Akt/mTor pathway;
- Figure 1, just up-regulated genes are shown, while providing an insight into down-regulated ones would be fine. The same Figure 1, panel H, shows a marked difference in Ddit3 levels between C and KPC, though it does not appear to reach the statistical significance... provided that protein levels increase as well, I would encourage the Authors to repeat the qPCR to check if indeed Ddit3 expression is not regulated at the mRNA level;
- the differences in muscle mass reported in Figure 2 appear very small. The easy suggestion is to use a different scale, to

render the results immediately appreciable to the reader. Are Xbp1^{fl/fl} and Xbp1^{mKO} (in the absence of the tumor) comparable in terms of body weight, muscle mass, etc.? The question arises since the differences detectable in Figure 2 A-C seem to parallel the preventive effects of Xbp1 ko. In other words, is Xbp1 ko not affecting at all UPR in non-tumor bearing mice?

- Figure 2F, it seems that the interfiber space is larger in the ko than in the fl/fl soleus, irrespective of the presence of the tumor. A staining to check for collagen deposition should be provided;

- Figure 4C, XBP1 expression in PBS injected Xbp1^{fl/fl} mice is virtually absent, while Figure 11 reports a detectable level; could this reflect the genetic manipulation? Could this impinge on the biological effects?

- data reported in Figure 8D show that the tumor mass is quite different in treated vs untreated mice, despite the lack of statistical significance. Along this line, tumors should be analyzed from the histological point of view, to verify that indeed, tumor cell number is reasonably unchanged.

Minor points

- the Authors frequently talk about 'myotube atrophy'; however, atrophy is a complex condition, which results from the interplay of several factors that could not be reflected by the microenvironment of a culture dish. 'Myotube reduction in size/dimension' should be used;

- page 13, the Authors talk about 'pro-cachectic' genes... genes are not cachectic, maybe their products could mediate cachexia;

- the quality of some western blotting (Figure 8I, as an example) is poor.

As a service to authors, EMBO provides authors with the possibility to transfer a manuscript that one journal cannot offer to publish to another EMBO publication. The full manuscript and if applicable, reviewers reports are automatically sent to the receiving journal to allow for fast handling and a prompt decision on your manuscript. For more details of this service, and to transfer your manuscript to another EMBO title please click on Link Not Available

RESPONSE TO REVIEWERS' COMMENTS

We are grateful to both the reviewers for their time spent reviewing our manuscript and providing excellent suggestions to improve the quality of the manuscript. We have now performed some additional experiments and analysis which further support the conclusions of our study. Our point-wise response to referees' comments is as follows:

REFEREE #1

(Comments on Novelty/Model System for Author):

The transgenic mice being utilized are optimal for these studies.

RESPONSE: We thank the reviewer for finding transgenic mice optimal for the study.

Referee #1 (Remarks for Author):

Targeting the canonical ER stress IRE1 α /XBP1 pathway counteracts pancreatic cancer-induced skeletal muscle wasting

This study has elucidated a novel pathway to address ER stress-associated muscle wasting in pancreatic cancer. Interestingly, this study demonstrates that the IRE1/XBP1 pathway is highly activated in the skeletal muscle of KPC tumor-bearing mice. Interestingly, this study identifies XBP1 as a key transcription factor that directly binds to the promoters of multiple cachexia-related genes, as well as its own promoter, in the KPC model, suggesting a positive feedback loop. Mechanistically, sXBP1 (spliced XBP1) enhances expression of genes involved in the ubiquitin-proteasome system (UPS) and autophagy, such as MAFbx, MuRF1, and LC3b. CHIP-qPCR revealed that sXBP1 binds to the promoter regions of these genes, thereby strengthening the hypothesis of this manuscript and indicating transcriptional regulation. Deletion or inhibition of XBP1 also suppressed components of the ER-associated degradation (ERAD) pathway and reduced activation of the proinflammatory IL-6-JAK-STAT3 signaling axis, further linking ER stress to cachectic signaling. Treatment with 4 μ 8C, an IRE1 endonuclease inhibitor, recapitulated the protective effects seen with Xbp1 knockout, attenuating proteolysis, autophagy, fatty acid oxidation, and STAT3 signaling. These findings position the IRE1/XBP1 axis as a central regulator of skeletal muscle wasting and a promising therapeutic target in PDAC-associated cachexia. Overall, the manuscript presents novel findings with strong supporting data, providing detailed mechanistic insights that implicate the IRE1/XBP1 pathway in cachexia. Appropriate controls have been included, and the manuscript is well-written, presenting novel findings with robust scientific rigor. Utilization of multiple transgenic mouse models is another strength. I have the following comments:

Major comments:

1. Figure 1 presents initial findings on ER stress using the gastrocnemius muscle. It would strengthen Figure 2 to include Laminin or H&E staining and cross-sectional area (CSA) measurements of the gastrocnemius muscle as well.

RESPONSE: We have used TA muscle which predominately contains glycolytic myofibers and soleus muscle which contains both glycolytic and oxidative myofibers for our histological analysis in **Figure 2**. Like soleus muscle, GA muscle also contains both glycolytic and oxidative myofibers. While we did not collect GA muscles for histological analysis, our results demonstrate that genetic ablation of XBP1 significantly inhibits the loss of GA muscle mass

along with TA and soleus muscle (**Figure 2E**) in response to KPC tumor growth. These results suggest that the inhibition of XBP1 attenuates KPC tumor-induced myofiber atrophy independent of muscle fiber type.

2. In Figure 2C, there is no change in muscle weight of TA and Soleus between *Xbp1*fl/fl and *Xbp1*mKO, but a significant decrease in myofiber CSA (Figure 2E-J), which is a marker of muscle mass loss. The authors should explain this discrepancy.

RESPONSE: This is an important point which was also raised by Referee # 3. There is a small decrease in muscle mass and average myofiber cross-sectional area (CSA) in the *Xbp1*^{mKO} mice compared with *Xbp1*^{fl/fl} mice in the absence of tumor. However, this difference is not significant when we perform two-way ANOVA to compare the differences in four groups. We have now performed another analysis in which the percentage decrease in muscle wet weight and average myofiber CSA in *Xbp1*^{fl/fl} and *Xbp1*^{mKO} mice in response to KPC tumor growth was evaluated. As shown in **new Figures 2E, 2I, and 2M**, there is a significant inhibition in the loss of muscle mass and average myofiber CSA in KPC tumor-bearing *Xbp1*^{mKO} mice compared to corresponding *Xbp1*^{fl/fl} mice. Similar analysis showed that pharmacological inhibition of IRE1 α /XBP1 axis using 4 μ 8C also significantly inhibits the loss of muscle wet weight and average myofiber CSA in response to tumor growth (**Figures 8C and 8G**)

3. In Figure 2A, there is no change in tumor-free body weight between *Xbp1*fl/fl and *Xbp1*mKO; body weight loss is a primary phenotype of cancer cachexia. Is this due to it being an early time point? Please include an explanation in the discussion.

RESPONSE: We did not find any significant difference between the tumor-free body weight of *Xbp1*^{fl/fl} and *Xbp1*^{mKO} mice by performing two-way ANOVA. This could be attributed to the fact that our mice are muscle-specific *Xbp1*-knockout mice which would not account for the changes that may have happened in other tissues, including fat content in response to KPC tumor growth. Our new analysis shows that the percentage reduction in the tumor-free body weight compared to their corresponding control (injected with PBS alone) mice is significantly less in KPC tumor-bearing *Xbp1*^{mKO} mice compared to KPC tumor-bearing *Xbp1*^{fl/fl} mice (**Figure 2D**). Furthermore, we found a significant increase in grip strength of *Xbp1*^{mKO} compared with *Xbp1*^{fl/fl} mice normalized by tumor-free body weight which further suggests that KPC tumor-induced muscle wasting is inhibited in *Xbp1*^{mKO} mice. We believe this new analysis about the decrease in tumor-free body weight and individual hind limb muscle weight makes the muscle phenotype readily apparent.

4. In Figure 3 J and L, why is there a difference in the protein expression of sXBP1 in the vehicle groups? Perhaps somewhat comparable exposures for vehicle controls would be useful.

RESPONSE: This was because the proteins were loaded on two different gels, and the exposure time was different for each blot. We have now run protein samples on the same gel which shows levels of sXBP1 protein in vehicle, 4 μ 8c, and IXA4 groups (**new Figure 3J, K**).

5. Including tumor volume data in the study would enhance its comprehensiveness.

RESPONSE. The KPC is an orthotopic model in which the cells were injected into the pancreas of the mice. It is not possible to accurately measure the tumor volume during the study. Since we did not find any difference in the wet weight of tumors between *Xbp1*^{fl/fl} and *Xbp1*^{mKO} mice, we did not measure the tumor volume after euthanizing the mice. We also found that there was no

significant difference in the wet weight of tumor between vehicle and 4 μ 8C-treated mice. However, based on the suggestion by the other Referee, we have now analyzed the tumor samples which showed comparable histology and number of Ki67⁺ cells in vehicle and 4 μ 8C-treated mice (**Supplemental Figure S5G-I**).

6. Figure 8I, blot quality is not good specifically p-STAT3. I would recommend changing these blots with clear ones.

RESPONSE: We have now repeated these western blots and included better quality blots.

7. Previous studies have demonstrated that inducing ER stress can be beneficial in eradicating PDAC tumors by chemotherapy, and tumor cells have mechanisms to overcome ER stress (PMID: 28811332, 32103170). The authors should briefly describe the tumor-centric impact of the inhibitor and cite these articles in the context of systemic ER stress inhibition.

RESPONSE: This is a very good suggestion. We have now discussed tumor-specific role of the ER stress in the “Discussion” section of this revised submission (Page # 21, highlighted text).

Minor comments:

1. The text conversion to PDF created some erroneous squares in the text in the introduction, line 7 on page 3, the last sentence on page 4, page 24, and page 25. Please correct.

RESPONSE: Our Word file has no such issues. This may have happened during the conversion to PDF at the manuscript submission site. We hope such issues won't be there in this revised submission.

2. Reference 17 states intramyocellular lipid droplet accumulation in cachexia. This does not fit with the sentence "Indeed, depletion of intramyocellular lipids, due to accelerated fatty acid metabolism, is a characteristic feature of the cachectic muscles of animal models of cancer cachexia (16-18)." Please change the description or the reference.

RESPONSE: This sentence was not accurate. It has been removed from the “Introduction” section.

3. I would recommend mentioning the physiological role/status of IRE1 α and XBP1 in the introduction.

RESPONSE: The physiological role of IRE1 α /XBP1 axis in skeletal muscle has now been mentioned in the “Introduction” section (Page # 4, highlighted text).

4. In the abstract, the sentence "Transcriptionally active XBP1 protein binds to the promoter region of multiple genes whose products are involved in skeletal muscle wasting", would be clearer if key target genes name were specified.

RESPONSE: We have now mentioned the gene names in the Abstract. Thank you!

REFEREE #3

The present study by Joshi and collaborators is aimed to investigate the relevance of the IRE1 α /XBP1 axis to the onset and progression of cachexia induced by an implantable model of pancreatic cancer (KPC). The Authors previously showed that ER stress contributes to cancer-

induced cachexia (PMID: 27206451) and that the IRE1 α /XBP1 axis is overactivated in the muscle of mice bearing the Lewis Lung Carcinoma (PMID: 31138662). The study by Joshi et al. extends the focus to cachexia induced by the KPC tumor. Most of the conclusions refer to results obtained by the RNAseq analysis performed on the gastrocnemius muscle. The hypothesis that the IRE1 α /XBP1 axis could play a causative role in KPC-induced cachexia is supported, partially at least, by experiments in which the axis is modulated by means of genetic or pharmacological tools.

Points to clarify/integrate/improve:

- the data showing that markers of protein turnover are up-regulated at both mRNA and protein levels just suggest that the system is altered. A dynamic assessment of protein turnover rates should be provided; even if not totally explanatory, measuring the autophagic flux and protein synthesis by puromycin incorporation will give an acceptable estimate.

RESPONSE: We have now performed additional experiments which suggest that protein turnover is affected. Specifically, in **Figure 11**, we demonstrate that the levels of ubiquitin-conjugated proteins and ratio of LC3bII/I are significantly elevated in the GA muscle of KPC tumor-bearing mice suggesting perturbations in both ubiquitin-proteasome system and autophagy. We have also performed similar studies using cultured primary myotubes. Our results demonstrate that treatment of myotubes with KPC conditioned medium (KPC-CM) causes significant inhibition in the puromycin incorporation (i.e., protein synthesis) in myotubes. Similar to the in vivo results, we found that treatment with KPC-CM increases amounts of ubiquitin-conjugated proteins, autophagy flux, and levels of IRE1 α and XBP1 in cultured myotubes (New **Figure S1** in the Supplemental data file).

- the Authors show that the IRE1 α /XBP1 axis is able to regulate STAT3 signaling in the muscle of tumor-bearing mice. The anatomical compartment (the muscle) is the only novel issue in this regard, since IRE1 α /XBP1-dependent activation of STAT3 was previously demonstrated in cancer cells (PMID: 28222747; PMID: 30214606).

RESPONSE. We appreciate the reviewer's comment bringing our attention to these published articles about the role of IRE1 α /XBP1 in the regulation of STAT3 signaling in melanoma and hepatocellular carcinoma. These published articles further support our findings about the role of IRE1 α /XBP1 in STAT3 signaling in cachectic skeletal muscle. However, our findings about the role of IRE1 α /XBP1 in the regulation of STAT signaling were unbiased and solely based on the preliminary findings by the RNA-Seq experiment which we confirmed by biochemical experiments and ChIP assay. It is important to note that while the promoter region of *Stat3* also contains a consensus sXBP1 binding site, we could not detect enrichment of sXBP1 to the promoter region of *Stat3* gene. Rather, we found that sXBP1 binds to the promoter region of *Il6* gene (**Figure 7G**). It is noteworthy that in addition to *IL6* and its own expression, our results demonstrate that sXBP1 directly regulates the gene expression of other molecules (e.g., *MAFbx*, *LC3B*, *ATG5*, and *Pdk4*) involved in muscle proteolysis in response to KPC tumor growth.

- page 11, the Authors state: The GSEA of RNA-seq dataset showed that the fatty acid metabolism was significantly increased. This sentence is totally misleading, since RNAseq data do not demonstrate a metabolic activity. This example very well fits with the overall impression

reported above, that many conclusions are drawn on the basis of mRNA expression data, while little insight into biological activity is provided.

RESPONSE: Our analysis of RNA-Seq dataset showed that the gene expression of many molecules involved in fatty acid metabolism/oxidation was affected by the deletion of XBP1 in skeletal muscle of tumor-bearing mice. We validated these findings by performing independent qPCR for important genes involved in this process. We have also now used a commercially available fatty acid oxidation activity assay kit (for medium chain fatty acids) to measure the differences in the two groups. Our new results demonstrate that there was a significant upregulation in the fatty acid oxidation in skeletal muscle of mice in response to tumor growth and there was a significant inhibition in the fatty acid oxidation in the skeletal muscle of KPC tumor-bearing *Xbp1^{mKO}* mice compared to corresponding *Xbp1^{fl/fl}* mice (**Figure 6E**). Moreover, our new results also demonstrate that the inhibition of IRE1 α /XBP1 axis reduces the amount of carbonylated (irreversibly oxidized) proteins in the skeletal muscle of tumor-bearing mice (**Figure 6F, G** and **Figure 8K, L**). These new results further support our RNA-seq analysis and qPCR results that the inhibition of IRE1/XBP1 reduces fatty acid oxidation in the skeletal muscle of KPC tumor-bearing mice.

- surprisingly, the muscles of KPC hosts (*Xbp1^{fl/fl}*) show a low intramyocellular lipid content (Oil-Red-O positive lipid droplets). Such a pattern is reverted when the KPC tumor is implanted into *Xbp1^{mKO}*. This is somewhat in contrast with other data in the literature (PMID: 39010842; PMID: 21766057), also in cancer patients (PMID: 38725139; PMID: 36189019; PMID: 35220502, among others);

RESPONSE: We agree with the reviewer. We had performed an oil-red O staining which was of poor quality. Reexamination of all the slides showed that there was non-specific background staining in many slides with oil-red-O which compounded the analysis/quantification. To specifically visualize lipid droplets within skeletal muscle, we have now performed immunostaining for Perilipin 2 (*Plin2*) protein, a widely used marker to identify lipid droplets in mammalian cells, including skeletal muscle. This experiment clearly showed that lipid droplets significantly increased in the TA muscle sections of KPC tumor-bearing mice compared to control mice. Furthermore, we found that there was a small, but significant reduction in Perilipin 2 staining intensity in the TA muscle sections of KPC tumor-bearing *Xbp1^{mKO}* mice compared with corresponding *Xbp1^{fl/fl}* mice (**Figure 6H, I**). This reduction in lipid content could be attributed to reduced uptake of fatty acid because the gene expression of CD36 (a membrane glycoprotein which facilitates fatty acid translocation across the cell membrane into the cytoplasm) is reduced in skeletal muscle of *Xbp1^{mKO}* mice (**Figure 6D**).

- page 17, the Authors define ERAD as another proteolytic pathway, which is not totally correct. Indeed, proteasome and autophagy are the main actors involved in ERAD. By contrast, ERAD is activated by peculiar mechanisms, which can make the difference with respect to the 'canonical' FoxO/Akt/mTor pathway.

RESPONSE: We completely agree with the reviewer. We have modified the text in the “Discussion” section to emphasize that ERAD pathway removes misfolded proteins through the ubiquitin-proteasome system and autophagy (highlighted text on Page # 18).

- Figure 1, just up-regulated genes are shown, while providing an insight into down-regulated ones would be fine. The same Figure 1, panel H, shows a marked difference in *Ddit3* levels

between C and KPC, though it does not appear to reach the statistical significance... provided that protein levels increase as well, I would encourage the Authors to repeat the qPCR to check if indeed Ddit3 expression is not regulated at the mRNA level.

RESPONSE: This is an excellent suggestion. We have now included down-regulated genes as well (**Figure 1F**). Furthermore, we have repeated qPCR analysis for Ddit3 (i.e., CHOP) which demonstrates that mRNA levels of Ddit3 are also significantly up-regulated in skeletal muscle of KPC tumor-bearing mice. These results suggest increased gene expression of Ddit3 in skeletal muscle of tumor-bearing mice.

- the differences in muscle mass reported in Figure 2 appear very small. The easy suggestion is to use a different scale, to render the results immediately appreciable to the reader. Are Xbp1^{fl/fl} and Xbp1^{mKO} (in the absence of the tumor) comparable in terms of body weight, muscle mass, etc.? The question arises since the differences detectable in Figure 2 A-C seem to parallel with the preventive effects of Xbp1 ko. In other words, is Xbp1 ko not affecting at all UPR in non-tumor bearing mice?

RESPONSE: Similar comment was also made by Referee # 1 which is addressed above. As presented in **Figure 2A** and **Figure S2B**, there is no statistically significant difference in the body weight of Xbp1^{fl/fl} and Xbp1^{mKO} mice even though there is a trend towards decreased body weight of Xbp1^{mKO} mice compared to Xbp1^{fl/fl} mice. We have also found that there is a small reduction in muscle wet weight and average myofiber CSA in Xbp1^{mKO} mice compared to Xbp1^{fl/fl} mice in the absence of tumor. These differences are statistically significant only when comparisons are made between control (i.e., without tumor) Xbp1^{fl/fl} and Xbp1^{mKO} mice using unpaired Student *t* test. However, no statistically significant differences were observed when we performed two-way ANOVA using all four groups. In addition to presenting the changes in wet weight of individual muscles and average myofiber CSA in four groups, we have also now included data that shows percentage change/decrease in the wet weight of individual hind limb muscles and average myofiber CSA that occurs in response to tumor growth. These results clearly show that there is a significant reduction in the loss of tumor-free body weight and wet weight of TA, GA, and soleus muscle of Xbp1^{mKO} mice compared with Xbp1^{fl/fl} mice in response to tumor growth (new **Figures 2D** and **2E**). Similar tumor-induced reduction in the average myofiber CSA in TA and soleus muscle in Xbp1^{mKO} mice was significantly less compared with Xbp1^{fl/fl} mice (new **Figure 2I** and **2M**). We believe this new analysis makes the phenotype readily appreciable.

We would also like to mention that even though there is no apparent difference in muscle phenotype in non-tumor bearing Xbp1^{fl/fl} and Xbp1^{mKO} mice, we observed some changes in the markers of UPR in skeletal muscle. For example, deletion of XBP1 led to an increase in the levels of IRE1 α and PERK protein in skeletal muscle of non-tumor bearing mice (**Supplemental Figure S4A, B**). There is also a reduction in the mRNA levels of various components of ERAD in skeletal muscle of Xbp1^{mKO} mice compared with Xbp1^{fl/fl} mice (**Figure S4C**) suggesting that targeted deletion of XBP1 affects UPR in skeletal muscle of non-tumor bearing mice.

- Figure 2F, it seems that the interfiber space is larger in the ko than in the fl/fl soleus, irrespective of the presence of the tumor. A staining to check for collagen deposition should be provided.

RESPONSE: By carefully examining all the sections, we confirm that intermyofiber space is comparable between Xbp1^{mKO} and Xbp1^{fl/fl} mice. We have now replaced the previous image with

a more representative image. In addition, we have included the whole images of laminin-stained TA and soleus muscle sections (from 3 mice in each group) which show no differences in intermyofiber space (supplemental **Figure S2C** and **S2D**). We also performed Sirius red staining which showed no apparent difference in the collagen deposition in the tumor-bearing *Xbp1^{fl/fl}* and *Xbp1^{mKO}* mice (**Figure S2F**).

- Figure 4C, XBP1 expression in PBS injected *Xbp1^{fl/fl}* mice is virtually absent, while Figure 1I reports a detectable level; could this reflect the genetic manipulation? Could this impinge on the biological effects?

RESPONSE: We have now included higher exposure of sXBP1 blot which shows presence of sXBP1 protein in PBS-injected *Xbp1^{fl/fl}* mice as well (**Figure 4C**).

- data reported in Figure 8D show that the tumor mass is quite different in treated vs untreated mice, despite the lack of statistical significance. Along this line, tumors should be analyzed from the histological point of view, to verify that indeed, tumor cell number is reasonably unchanged.

RESPONSE: Based on the reviewer's suggestion, we have now performed histological analysis of tumor samples. Our results show that tumor histology (by performing H&E staining) was comparable between vehicle and 4 μ 8C groups (Supplemental **Figure S5G**). Furthermore, immunohistochemical analysis showed that there was no significant difference in the number of Ki67⁺ cells between the two groups (Supplemental **Figure S5H, I**).

Minor points

- the Authors frequently talk about 'myotube atrophy'; however, atrophy is a complex condition, which results from the interplay of several factors that could not be reflected by the microenvironment of a culture dish. 'Myotube reduction in size/dimension' should be used.

RESPONSE: We agree with the reviewer. We have made this correction throughout the manuscript.

- page 13, the Authors talk about 'pro-cachectic' genes... genes are not cachectic, maybe their products could mediate cachexia.

RESPONSE: This is absolutely true. We have now made this correction in the revised manuscript.

- the quality of some western blotting (Figure 8I, as an example) is poor.

RESPONSE: We have now re-run the samples and included new blots. We hope that the reviewer will find new blots of publication quality.

20th Aug 2025

Dear Prof. Kumar,

Thank you for the submission of your revised manuscript to EMBO Molecular Medicine. We have now heard back from the two referees who agreed to re-evaluate your manuscript. As you will see from their reports below, while referee #1 recommends publication of the revised manuscript, referee #3 remains critical regarding the protein turnover and the novelty. We concluded that raised concerns are justified and should be addressed in a final round of major revision. Please perform additional experiments to address point #1 as suggested by the referee #3 and point #2 should be addressed by better highlighting the conceptual advance of the study compared to previous publications that should be appropriately cited. In addition, for the next submission please adhere to the journal's formatting requirements. Please check our Author Guidelines for more information <https://www.embopress.org/page/journal/17574684/authorguide#manuscriptpreparation>

Further consideration of a revision that addresses reviewer's concerns in full will entail an additional round of review. Acceptance or rejection of the manuscript will depend on the completeness of your responses included in the next, final version of the manuscript. For this reason, and to save you from any frustrations in the end, I would strongly advise against returning an incomplete revision.

We would welcome the submission of a revised version within three months for further consideration. Please let us know if you require longer to complete the revision.

I look forward to receiving your revised manuscript.

Yours sincerely,

Zeljko Durdevic

Zeljko Durdevic
Senior Editor
EMBO Molecular Medicine

We require:

- 1) A .docx formatted version of the manuscript text (including legends for main figures, EV figures and tables). Please make sure that the changes are highlighted to be clearly visible.
- 2) Individual production quality figure files as .eps, .tif, .jpg (one file per figure). For guidance, download the 'Figure Guide PDF': (<https://www.embopress.org/page/journal/17574684/authorguide#figureformat>).
- 3) A .docx formatted letter INCLUDING the reviewers' reports and your detailed point-by-point responses to their comments. As part of the EMBO Press transparent editorial process, the point-by-point response is part of the Review Process File (RPF), which will be published alongside your paper.
- 4) A complete author checklist, which you can download from our author guidelines (<https://www.embopress.org/page/journal/17574684/authorguide#submissionofrevisions>). Please insert information in the checklist that is also reflected in the manuscript. The completed author checklist will also be part of the RPF.
- 5) Please note that all corresponding authors are required to supply an ORCID ID for their name upon submission of a revised

manuscript.

6) It is mandatory to include a 'Data Availability' section after the Materials and Methods. Before submitting your revision, primary datasets produced in this study need to be deposited in an appropriate public database, and the accession numbers and database listed under 'Data Availability'. Please remember to provide a reviewer password if the datasets are not yet public (see <https://www.embopress.org/page/journal/17574684/authorguide#dataavailability>).

.

- the medical issue you are addressing,

- the results obtained and

- their clinical impact.

12) Author contributions: You will be asked to provide CRediT (Contributor Role Taxonomy) terms in the submission system. These replace a narrative author contribution section in the manuscript.

13) A Conflict of Interest statement should be provided in the main text.

14) Every published paper now includes a 'Synopsis' to further enhance discoverability. Synopses are displayed on the journal webpage and are freely accessible to all readers. They include a short stand first (maximum of 300 characters, including space)

as well as 2-5 one-sentences bullet points that summarizes the paper. Please write the bullet points to summarize the key NEW findings. They should be designed to be complementary to the abstract - i.e. not repeat the same text. We encourage inclusion of key acronyms and quantitative information (maximum of 30 words / bullet point). Please use the passive voice. Please attach these in a separate file or send them by email, we will incorporate them accordingly.

15) Include a Reagents and Tools Table as part of the Methods section, which can be downloaded from our author guidelines (<https://www.embopress.org/page/journal/17574684/authorguide#structuredmethods>)

***** Reviewer's comments *****

Referee #1 (Comments on Novelty/Model System for Author):

The transgenic mouse models are optimal.

Referee #1 (Remarks for Author):

The manuscript provides novel insights into the mechanisms of cachexia. The models utilized are very rigorous, and all the experiments are well-controlled. All of my concerns have been adequately addressed. The findings are highly significant and likely to strongly advance the field forward. I recommend accepting the manuscript.

Referee #3 (Remarks for Author):

In the present version of the study, Joshi and collaborators satisfactorily addressed many of the points raised during the first revision round. Still, a couple of aspects could have been better dealt with, at least by discussing the choices done by the Authors:

as for protein turnover, the Authors added some data, such as the levels of ubiquitylated proteins and the LC3BI/II ratio, as indirect measures of ubiquitin-dependent proteolysis and autophagy. In addition, they performed in vitro experiments using C2C12 myotubes exposed to the KPC conditioned medium, to demonstrate reduced puromycin incorporation, increased levels of ubiquitylated proteins and of the autophagic flux. Those data are partially corroborating the hypothesis of increased protein turnover, however, puromycin incorporation and autophagic flux in the living KPC hosts would have been more informative;

the Author reply to the 'de facto' lack of novelty about IRE1 α /XBP1 in STAT3 signaling does not really addresses the issue. The point should be acknowledged in the paper.

RESPONSE TO REVIEWER'S COMMENTS

We thank both reviewers for finding the revised version of our manuscript significantly improved and acceptable for publication. Reviewer #3 provided two additional comments, which we have addressed in this response. All changes made to the manuscript in response to the reviewer's comments are highlighted in blue font.

Reviewer # 3

REVIEWER: In the present version of the study, Joshi and collaborators satisfactorily addressed many of the points raised during the first revision round. Still, a couple of aspects could have been better dealt with, at least by discussing the choices done by the Authors:

as for protein turnover, the Authors added some data, such as the levels of ubiquitylated proteins and the LC3BI/II ratio, as indirect measures of ubiquitin-dependent proteolysis and autophagy. In addition, they performed in vitro experiments using C2C12 myotubes exposed to the KPC conditioned medium, to demonstrate reduced puromycin incorporation, increased levels of ubiquitylated proteins and of the autophagic flux. Those data are partially corroborating the hypothesis of increased protein turnover, however, puromycin incorporation and autophagic flux in the living KPC hosts would have been more informative;

OUR RESPONSE: We have now conducted an additional experiment to measure autophagy flux in KPC tumor-bearing mice, with the results presented in **Figure EV1A and B**. In addition, we performed a puromycin incorporation assay to assess protein synthesis in vivo. Although protein synthesis is generally thought to decline in skeletal muscle during cancer progression, some studies have reported that it remains unchanged or even increases in animal models and patients with cancer cachexia (PMID: 36864755). Consistent with these observations, we found a modest but statistically significant increase in puromycin incorporation in the skeletal muscle of KPC tumor-bearing mice (**Figure EV1C and D**).

Interestingly, our in vitro studies also showed an initial increase in puromycin incorporation in cultured myotubes treated with conditioned medium from KPC cells (KPC-CM). However, after 24 hours of incubation with KPC-CM, puromycin incorporation declined, suggesting that the rate of protein synthesis may decrease only at later stages of tumor progression. The mechanisms underlying the observed increase in protein synthesis in skeletal muscle in response to KPC tumor burden remain unclear. However, our results are consistent with a recent human study reporting enhanced markers of translation initiation in the skeletal muscle of pancreatic cancer patients with cachexia (PMID: 40931079). Moreover, in another recent study from our group using single-nucleus RNA sequencing, we found that markers of translation initiation and ribosome biogenesis are elevated specifically in type IIb myofibers of KPC tumor-bearing mice (bioRxiv: <https://doi.org/10.1101/2025.09.15.676415>)

We propose that this increase in protein synthesis may represent a compensatory mechanism aimed at mitigating excessive muscle loss during tumor progression. Nonetheless, our findings collectively support the conclusion that proteostasis is disrupted in the skeletal muscle of KPC tumor-bearing mice.

REVIEWER: the Author reply to the 'de facto' lack of novelty about IRE1 α /XBP1 in STAT3 signaling does not really addresses the issue. The point should be acknowledged in the paper.

OUR RESPONSE: We have now discussed and cited the two articles about the role of XBP1 in the regulation of IL-6 expression in melanoma and hepatocellular carcinoma (Page # 19, highlighted text and **reference # 64 and 65**).

17th Oct 2025

Dear Prof. Kumar,

Thank you for the submission of your revised manuscript to EMBO Molecular Medicine. I am pleased to inform you that we will be able to accept your manuscript pending the following final amendments:

1) Figures: Image resolution for blots in the figure set and in the source data are of low resolution and data size. This is a common result of converting original 16-bit TIFF images to RGB format for publication, and while not a cause for concern, it can sometimes give the impression of image alteration to critical readers. To resolve this, please upload the blots in the figure set at a higher resolution. Please check "Author Guidelines" for more information.

<https://www.embopress.org/page/journal/17574684/authorguide#expandedview>

2) In the main manuscript file, please do the following:

- Please address all comments suggested by our data editors listed below:

o Figure legends:

1. Please note that the exact p values are not provided in the legends of figures 3B, I; 4B, D; 5D, 6D, 8F, H, J; EV1 D, L.

2. Please indicate the statistical test used for data analysis in the legends of figures 1E, F; 2F; 4A.

3. Please note that information related to n is missing in the legend of figure 1E.

4. Please note that the error bars are not defined in the legend of figure 2F.

- Remove "Conflict of interest" from the title page.

- In Methods, provide the antibody dilutions that were used for each antibody.

- Please remove Reagents and Tools Table from the manuscript file and uploaded it as a separate file. More information on how to adhere to this format as well as downloadable templates (.docx) for the Reagents and Tools Table can be found in our author guidelines: <https://www.embopress.org/page/journal/17574684/authorguide#structuredmethods>

An example of a paper with Structured Methods can be found here:

<https://www.embopress.org/doi/full/10.1038/s44320-024-00037-6#sec-4>

- Rename "Declaration of interest" to "Disclosure and competing interests statement". We updated our journal's competing interests policy in January 2022 and request authors to consider both actual and perceived competing interests. Please review the policy <https://www.embopress.org/competing-interests> and update your competing interests if necessary.

- Author contributions: Please remove it from the manuscript and specify author contributions in our submission system. CRediT has replaced the traditional author contributions section because it offers a systematic machine-readable author contributions format that allows for more effective research assessment. You are encouraged to use the free text boxes beneath each contributing author's name to add specific details on the author's contribution. More information is available in our guide to authors:

<https://www.embopress.org/page/journal/17574684/authorguide#authorshipguidelines>

- Indicate in legends number and nature of replicates and exact p= values, not a range, along with the statistical test used. To keep the figures "clear" some authors found providing an Appendix table Sx with all exact p-values preferable. You are welcome to do this if you want to.

- In data availability remove the sentence "All other raw data generated for this study has been included with this manuscript."

- Correct the reference citation in the text and reference list. In the text a reference should be cited by author and year of publication. Include a space between a word and the opening parenthesis of the reference that follows. In the reference list, citations should be listed in alphabetical order. Where there are more than 10 authors on a paper, 10 will be listed, followed by "et al.". Also, please remove DOIs. Please check "Author Guidelines" for more information.

<https://www.embopress.org/page/journal/17574684/authorguide#referencesformat>

3) Tables: Please remove Tables EV1 and EV2 and their legends from the manuscript, upload them as separate files.

4) The Paper Explained: Please provide "The Paper Explained" and add it to the main manuscript text. Please check "Author Guidelines" for more information. <https://www.embopress.org/page/journal/17574684/authorguide#researcharticleguide>

5) Synopsis:

- Synopsis image: Please resize the image to 550 px-wide x 300-600 pixels high and upload it as a high-resolution jpeg file.

6) As part of the EMBO Publications transparent editorial process initiative (see our Editorial at

<http://embomolmed.embopress.org/content/2/9/329>), EMBO Molecular Medicine will publish online a Review Process File (RPF) to accompany accepted manuscripts. This file will be published in conjunction with your paper and will include the anonymous referee reports, your point-by-point response and all pertinent correspondence relating to the manuscript. Let us know whether you agree with the publication of the RPF and as here, if you want to remove or not any figures from it prior to publication. Please note that the Authors checklist will be published at the end of the RPF.

7) Please provide a point-by-point letter INCLUDING my comments as well as the reviewer's reports and your detailed responses (as Word file).

I look forward to reading a new revised version of your manuscript as soon as possible.

Yours sincerely,

Zeljko Durdevic

Zeljko Durdevic
Senior Editor
EMBO Molecular Medicine

*** Instructions to submit your revised manuscript ***

1) a .docx formatted version of the manuscript text (including Figure legends and tables)

2) Separate figure files*

3) supplemental information as Expanded View and/or Appendix. Please carefully check the authors guidelines for formatting Expanded view and Appendix figures and tables at <https://www.embopress.org/page/journal/17574684/authorguide#expandedview>

4) a letter INCLUDING the reviewer's reports and your detailed responses to their comments (as Word file).

5) The paper explained: EMBO Molecular Medicine articles are accompanied by a summary of the articles to emphasize the major findings in the paper and their medical implications for the non-specialist reader. Please provide a draft summary of your article highlighting

This may be edited to ensure that readers understand the significance and context of the research.

Please refer to any of our published articles for an example.

6) Author contributions: the contribution of every author must be detailed in a separate section.

7) EMBO Molecular Medicine now requires a complete author checklist (<https://www.embopress.org/page/journal/17574684/authorguide>) to be submitted with all revised manuscripts. Please use the checklist as guideline for the sort of information we need WITHIN the manuscript. The checklist should only be filled with page numbers where the information can be found. This is particularly important for animal reporting, antibody dilutions (missing) and exact values and n that should be indicated instead of a range.

8) Every published paper now includes a 'Synopsis' to further enhance discoverability. Synopses are displayed on the journal webpage and are freely accessible to all readers. They include a short stand first (maximum of 300 characters, including space) as well as 2-5 one sentence bullet points that summarise the paper. Please write the bullet points to summarise the key NEW findings. They should be designed to be complementary to the abstract - i.e. not repeat the same text. We encourage inclusion of key acronyms and quantitative information (maximum of 30 words / bullet point). Please use the passive voice. Please attach these in a separate file or send them by email, we will incorporate them accordingly.

You are also welcome to suggest a striking image or visual abstract to illustrate your article. If you do please provide a jpeg file 550 px-wide x 300-600px high.

9) A Conflict of Interest statement should be provided in the main text

10) Please note that we now mandate that all corresponding authors list an ORCID digital identifier. This takes <90 seconds to complete. We encourage all authors to supply an ORCID identifier, which will be linked to their name for unambiguous name identification.

Currently, our records indicate that the ORCID for your account is 0000-0001-8571-2848.

Link Not Available

11) Include a Reagents and Tools Table as part of the Methods section, which can be downloaded from our author guidelines (<https://www.embopress.org/page/journal/17574684/authorguide#structuredmethods>)

Photos 400-800 DPI

*Additional important information regarding figures and illustrations can be found at

<https://bit.ly/EMBOPressFigurePreparationGuideline>. See also figure legend preparation guidelines:

<https://www.embopress.org/page/journal/17574684/authorguide#figureformat>

***** Reviewer's comments *****

Referee #3 (Remarks for Author):

The authors satisfactorily addressed the unclear points

The authors addressed the remaining editorial issues.

31st Oct 2025

Dear Prof. Kumar,

We are pleased to inform you that your manuscript is accepted for publication and is now being sent to our publisher to be included in the next available issue of EMBO Molecular Medicine.

Zeljko Durdevic
Senior Editor
EMBO Molecular Medicine
